# Progress in Gallium Oxide Field-Effect Transistors for High-Power and RF Applications

**DOI:** 10.3390/ma16247693

**Published:** 2023-12-18

**Authors:** Ory Maimon, Qiliang Li

**Affiliations:** 1Department of Electrical Engineering, George Mason University, Fairfax, VA 22030, USA; omaimon@gmu.edu; 2Nanoscale Device and Characterization Division, National Institute of Standards and Technology, Gaithersburg, MD 20899, USA; 3Quantum Science & Engineering Center, George Mason University, Fairfax, VA 22030, USA

**Keywords:** gallium oxide, wide-bandgap semiconductor, field-effect transistors (FETs), high power, RF, defects

## Abstract

Power electronics are becoming increasingly more important, as electrical energy constitutes 40% of the total primary energy usage in the USA and is expected to grow rapidly with the emergence of electric vehicles, renewable energy generation, and energy storage. New materials that are better suited for high-power applications are needed as the Si material limit is reached. Beta-phase gallium oxide (β-Ga_2_O_3_) is a promising ultra-wide-bandgap (UWBG) semiconductor for high-power and RF electronics due to its bandgap of 4.9 eV, large theoretical breakdown electric field of 8 MV cm^−1^, and Baliga figure of merit of 3300, 3–10 times larger than that of SiC and GaN. Moreover, β-Ga_2_O_3_ is the only WBG material that can be grown from melt, making large, high-quality, dopable substrates at low costs feasible. Significant efforts in the high-quality epitaxial growth of β-Ga_2_O_3_ and β-(Al_x_Ga_1−x_)_2_O_3_ heterostructures has led to high-performance devices for high-power and RF applications. In this report, we provide a comprehensive summary of the progress in β-Ga_2_O_3_ field-effect transistors (FETs) including a variety of transistor designs, channel materials, ohmic contact formations and improvements, gate dielectrics, and fabrication processes. Additionally, novel structures proposed through simulations and not yet realized in β-Ga_2_O_3_ are presented. Main issues such as defect characterization methods and relevant material preparation, thermal studies and management, and the lack of p-type doping with investigated alternatives are also discussed. Finally, major strategies and outlooks for commercial use will be outlined.

## 1. Introduction

The power semiconductor market observed a 30% growth in 2022, and continual growth is expected as more electrical energy passes through power electronics, approximately 30% in 2019 and up to 80% in the next decade [1,2]. High-power semiconductor applications are classified into high-power (low-frequency) or high-frequency, RF. As silicon power devices reach their limit at breakdown voltages up to 6.5 kV and have a high temperature capability up to 200 °C [3], wide-bandgap (WBG) materials offer improved efficiency, large power ratings, high switching speeds, and RF performance. While SiC and GaN have been the dominant WBG semiconductors with commercially available devices, ultra-wide-bandgap (UWBG) beta-phase gallium oxide (β-Ga_2_O_3_) is emerging as a material for next-generation high-power and RF electronics.

With its bandgap of 4.7–4.9 eV, large theoretical breakdown field of 8 MV cm^−1^, and high electron saturation velocity of 2 × 10^7^ cm s^−1^, β-Ga_2_O_3_ has a Baliga figure of merit (BFOM), indicating DC conduction losses, as well as a Johnson figure of merit (JFOM) for RF performance, higher than those of GaN and SiC [4,5,6,7]. Additionally, the ability to grow bulk substrates from the melt gives β-Ga_2_O_3_ a significant cost advantage over SiC and GaN [8]. However, difficult challenges face β-Ga_2_O_3_ in the form of a lack of shallow p-type dopants and low thermal conductivity, which is especially difficult for high-power applications where heat dissipation is essential. Heterostructures with p-type oxides have been fabricated with high performance; however, most of the research on β-Ga_2_O_3_ has focused on unipolar devices.

Great strides have been made in both high-power and RF β-Ga_2_O_3_ field-effect transistors (FETs) with continually improving material quality and fabrication processes. High-power lateral FETs have reported breakdown voltages up to 10 kV and BFOMs near 1 GW cm^−2^, while vertical devices have yet to achieve similar performance. The factors limiting vertical FETs are largely the lack of p-type dopants, which minimizes current-blocking capabilities, gate dielectric quality, stability, and robustness [9]. Many of the FET structural and material improvements, discussed in Section 3, that have been tested on lateral FETs can similarly be applied to vertical devices. Most β-Ga_2_O_3_ FETs are depletion-mode (D-mode), or normally on, due to the unipolar nature of β-Ga_2_O_3_ devices. The off-state leakage in D-mode FETs is more prominent than in enhancement-mode (E-mode), or normally off, FETs; however, their fabrication is more difficult and often requires band bending a heterointerface to deplete the existing channel. RF FETs are predominantly lateral devices with thin channel layers and highly scaled gate lengths for strong gate control and reduced parasitics. Techniques such as delta-doping and modulation doping are used to form a two-dimensional electron gas (2DEG) with a high carrier concentration and mobility. Maximum oscillating frequencies near 50 GHz with high breakdown electric fields have been reported, showing potential for high-power RF β-Ga_2_O_3_ FETs in the future. With the tremendous progress that has already been accomplished in the field, β-Ga_2_O_3_ presents itself as a strong candidate for high-power and high-frequency applications, but not without its challenges to overcome.

Previous review articles on β-Ga_2_O_3_ FETs have reported the chronological development in device design and performance [10], or focused specifically on RF FETs [7], E-mode FETs [11], or vertical GaN and β-Ga_2_O_3_ FETs [9]. Other review papers have covered FETs designed for both high-power and RF applications [12,13]. This review is formatted to aid current and future β-Ga_2_O_3_ high-power and RF FET researchers by separately discussing the different steps of FET fabrication, ranging from structures, materials, ohmic contacts, gate dielectrics, and material preparation. An overview of various material and FET defect characterization techniques is presented as well.

This article presents a comprehensive overview of the progress in β-Ga_2_O_3_ FETs, current challenges, and potential strategies to overcome them. Section 2 discusses the crystal structure and material properties, including FOM comparisons, bulk and epitaxial growth, and the doping of β-Ga_2_O_3_. Section 3 reviews many of the recent transistor designs for both high-power and RF applications. Section 3.1 focuses on structures that have been implemented, as well as proposed structures through technology computer-aided design (TCAD). Section 3.2 summarizes FETs with different channel and substrate materials such as semi-insulating homoepitaxial or heterostructure layers in the channel, and high thermally conductive substrates. Section 3.3 reviews metals and processes for high-quality ohmic contact formation and Section 3.4 gives an overview of different gate dielectrics used in β-Ga_2_O_3_ FETs. Section 4 discusses the importance of defect engineering, various characterization methods, and material preparation to improve interface quality. Section 5 gives a high-level overview of the current challenges and steps needed to add β-Ga_2_O_3_ devices to the market. Section 6 briefly summarizes the most promising applications and trends of β-Ga_2_O_3_ FETs. Section 7 then summarizes the advancements in β-Ga_2_O_3_ and provides an outlook of the future of β-Ga_2_O_3_.

## 2. Crystal Growth and Material Properties of β-Ga_2_O_3_

### 2.1. Different Phases

In 1952, Roy et al. discovered five polymorphs of Ga_2_O_3_ (α, β, γ, δ, and ε) using a gallia gel–water system, and determined that the β phase is the stable form [14]. Using first-principles calculations, Yoshioka et al. found that the theoretical formation energies of the different phases are in the order of β < ε < α < δ < γ, confirming that β-Ga_2_O_3_ is stable, while the other polymorphs show metastable behavior and transform into β-Ga_2_O_3_ at high temperatures [15]. In 2013, Playford et al. discovered another metastable phase (κ) via the thermal decomposition of Ga_5_O_7_(OH) above 500 °C [16]. The phase transitions compiled from Roy et al. and Playford et al. are shown in ref. [17].

The crystal structure of β-Ga_2_O_3_ is monoclinic and belongs to the C2/m space group with lattice constants of a = 12.2 Å, b = 3.0 Å, c = 5.8 Å, α = 90°, β = 104°, and γ = 90° (Figure 1a). The unique structure has two Ga sites, one with a tetrahedral geometry and one with an octahedral geometry, as well as three O sites, leading to high anisotropy in many of its material properties [18,19,20,21].

### 2.2. Material Properties

The band structure of β-Ga_2_O_3_, calculated using first-principles density functional theory (DFT) (Figure 1b), shows an indirect gap of 4.84 eV and a direct gap of 4.88 eV; however, β-Ga_2_O_3_ is largely considered a direct-gap semiconductor because of the close proximity of the gaps. The conduction band dispersion estimates an electron effective mass of ≈0.28 m_e_, where m_e_ is the rest electron mass. The valence band, however, exhibits almost no dispersion and, therefore, exhibits a very large hole effective mass due to the localized self-trapping of holes [19,23].

Experimentally observed bandgaps range between 4.7 and 4.9 eV [19,24], projecting a critical breakdown electric field, E_br_, of 6–8 MV cm^−1^. Various figures of merit (FOM), discussed below, have been developed to compare semiconductors for high-power applications. The Baliga FOM (BFOM) is an estimate of DC conduction losses in a material and is defined as both ε·µ·E_br_^3^, where ε is the material dielectric constant and µ is the carrier mobility, and as V_br_^2^ R_on,sp_^−1^ for devices, where V_br_ is the breakdown voltage and R_on,sp_ is the specific on-resistance. The theoretical BFOM of β-Ga_2_O_3_ is approximately 28 GW cm^−2^ and ≈3214 times larger than that of Si. Other power device metrics include the Johnson FOM (JFOM), which represents the power–frequency capability; the Baliga high-frequency FOM (BHFFOM), which is a measure of switching losses; the Keyes FOM for thermal capability for power density and speed; and the Huang chip area manufacturing FOM (HCAFOM) as an indicator of chip area requirements. The material properties and FOMs of β-Ga_2_O_3_ compared to other materials are summarized in Table 1 [6,12,25].

It is important to note the low and anisotropic thermal conductivity in β-Ga_2_O_3_ of 27.0 W m^−1^ K^−1^ in the [010] direction and 10.9 W m^−1^ K^−1^ in [100] [26]. The difference in thermal conductivity of [010] β-Ga_2_O_3_ and [100] β-Ga_2_O_3_ might not seem large compared to those of other (ultra)-wide-bandgap ((U)WBG) materials; however, simulations have shown that the max temperature rise in devices has a decreasing rate dependence on thermal conductivity, and that ≈105 °C and ≈61 °C max temperature rises are simulated for [100] and [010] β-Ga_2_O_3_, respectively. On the other hand, the simulated max temperature rises for SiC and diamond are ≈34 °C and ≈30 °C, respectively [27].

At low doping densities below 10^18^–10^19^, electron interactions with polar longitudinal optical (LO) phonons is identified as the dominant scattering mechanism, limiting the theoretical bulk mobility to ≤250 cm^2^ V^−1^ s^−1^, while at higher doping concentrations, impurity scattering is dominant [28,29,30]. Even though β-Ga_2_O_3_ has a low mobility, β-Ga_2_O_3_ maintains higher FOMs than GaN and SiC because of the square or cubic dependence on breakdown voltage and only a linear dependence on mobility.

**Table 1 materials-16-07693-t001:** Material properties and FOMs, relative to Si, of β-Ga_2_O_3_ compared to other semiconductors [6,12,31].

Material Properties	Si	GaAs	4H-SiC	GaN	β-Ga_2_O_3_	Diamond
Bandgap, E_g_ (eV)	1.1	1.4	3.3	3.4	4.9	5.5
Dielectric Constant, ε	11.8	12.9	9.7	9	10	5.5
Breakdown field, E_br_ (MV cm^−1^)	0.3	0.4	2.5	3.3	8	10
Electron mobility, µ (cm^2^ V^−1^ s^−1^)	1480	8400	1000	1250	200–250	2000
Saturation velocity, v_sat_ (10^7^ cm s^−1^)	1	1.2	2	2.5	1.8–2	1
Thermal conductivity, λ (W m^−1^ K^−1^)	150	55	270	210	10.9–27	1000
$\mathrm{BFOM}=\varepsilon_{r}\mu E_{br}^{3}$	1	14.7	317	846	3214	24,660
$\mathrm{JFOM}=E_{br}^{2}v_{s}^{2}/4\pi^{2}$	1	1.8	278	1089	2844	1100
$\mathrm{BHFFOM}=\mu E_{br}^{2}$	1	10.1	46.3	100.8	142.2	1501
$\mathrm{Keyes}\;\mathrm{FOM}={\lambda \left[\left(c{\cdot v}_{s}\right)/(4\pi \varepsilon )\right]}^{1/2}$	1	0.3	3.6	1.8	0.2	41.5
$\mathrm{HCAFOM}=\varepsilon \mu^{0.5}E_{br}^{2}$	1	5	48	85	279	619

### 2.3. Crystal Growth

One of the greatest advantages of β-Ga_2_O_3_ is the potential for ultra-low-cost, large-size (diameter 100–150 mm), high-quality substrates made possible via melt growth. β-Ga_2_O_3_ is the only WBG semiconductor that can be grown from the melt, and therefore, the cost of a β-Ga_2_O_3_ wafer is expected to be approximately 80% cheaper than that of SiC [8]. The different bulk crystal growth techniques are edge-defined film-fed growth (EFG) [32,33], Czochralski (CZ) [34], vertical Bridgman (VB) [35,36], floating zone (FZ) [37,38], and Verneuil [39,40]. From all the methods, EFG has so far grown larger-diameter substrates with high quality, low defect densities, and a relatively wide doping range [22,41].

### 2.4. Epitaxial Growth

The main methods of β-Ga_2_O_3_ thin film growth that have been developed include molecular beam epitaxy (MBE), plasma-assisted MBE (PAMBE), metal–organic chemical vapor deposition (MOCVD), halide vapor-phase epitaxy (HVPE), and low-pressure chemical vapor deposition (LPCVD). MBE has the advantage of growing very high-quality thin films with less impurities and precise control over the growth rate and doping (10^16^–10^20^ cm^−3^). It suffers, however, from low growth rates of 0.05–0.18 µm h^−1^ that make it impractical for thick epitaxial layers used in vertical devices, but ideal for lateral thin-channel devices. PAMBE uses an activated oxygen source to help the growth of β-Ga_2_O_3_ thin films, and has been shown to reduce background (unintentional) impurity concentrations [42,43,44]. MOCVD, also called metal–organic vapor-phase epitaxy (MOVPE), also produces high-purity thin films with controlled doping (10^17^–8 × 10^19^ cm^−3^) at higher growth rates of 0.8 µm h^−1^ and less costly than MBE, making MOCVD conducive for large-scale production. HVPE has a minimum doping concentration at the order of 10^15^ cm^−3^ and a significantly higher growth rate, with the maximum reported rate of 250 µm h^−1^. It is therefore used in thick epi layer growth for vertical devices [45]. The tradeoff for the higher growth rates in HVPE is lower-quality thin films with rougher surfaces and more defects. LPCVD is a scalable and lower-cost method that produces high-quality thin films with growth rates ranging from 0.5 to 10 µm h^−1^, controlled doping in the range of 10^17^–10^19^ cm^−3^, and a heterostructure capability [46,47]. LPCVD is the least used of the three growth techniques, but can provide a path for scalable, production-level β-Ga_2_O_3_ wafers. Additionally, MBE, MOCVD, and LPCVD can grow heterostructures, unlike HVPE. More detail on these growth methods can be found in refs. [17,48,49].

### 2.5. Doping Strategies

DFT calculations have been used to find the energy levels of various impurities, oxygen vacancies (V_O_), and gallium vacancies (V_Ga_) in the β-Ga_2_O_3_ bandgap. Oxygen vacancies act as deep donors more than 1 eV below the conduction band (E_C_), and gallium vacancies act as deep acceptors more than 1 eV above the valence band (E_V_) [50,51]. These vacancies do not contribute to conduction, but only act as doping compensation. Shallow donors found via DFT include Si_Ga(I)_ (Si impurity in Ga_I_ site), Ge_Ga(I)_, Sn_Ga(II)_, Cl_O(I)_, and F_O(I)_, with energy levels very near E_C_ [52]; however, the majority of experimentally used donors are Si, Sn, and Ge [53,54]. Acceptor impurities such as N, Sr, Zn, Cd, Ca, Be, Mg, and Fe all have levels more than 1.3 eV above E_V_, indicating that p-type doping is not possible, and is a major challenge in the development of β-Ga_2_O_3_ devices [50,55]. Deep acceptors are used to form highly resistive semi-insulating layers.

The donor levels of Si and Ge in MBE, LPCVD, CZ, and EFG samples, calculated using temperature-dependent Hall and conductivity measurements, ranged from 15 to 31 meV below E_C_, indicating shallow donors, while the Mg and Fe levels were located at E_C_—0.86 eV and E_C_—1.1 eV, respectively [53]. Mobility dependence on carrier concentration is expected to flatten at 250 cm^2^ V^−1^ s^−1^ as the carrier concentration approaches 10^15^ cm^−3^, and drops significantly as it increases above 10^17^ cm^−3^ (Figure 2) [31].

While p-type behavior is not available when using conventional methods, some groups have observed hole conduction when compensating donors are reduced [56]. The p-type conductivity of thin-film β-Ga_2_O_3_ by reducing the mean free path of carriers using amphoteric Zn doping was able to achieve ultra-high breakdown fields of 13.2 MV cm^−1^, beyond that of the theoretical β-Ga_2_O_3_ [57]. Another technique to modulate between high n-type and p-type conductivity was developed via controlled H incorporation, where p-type conductivity with an acceptor state 42 meV above E_V_ was observed after direct H diffusion, and n-type conductivity with a donor state 20 meV below E_C_ was observed after filling up oxygen vacancies by annealing in O_2_ [58].

## 3. β-Ga_2_O_3_ FET Designs

The following section reviews many of the current FET designs, including their structure, channel materials, substrate materials, ohmic contact formation, and gate dielectrics. Their process steps, use cases, advantages, and disadvantages are discussed. The tables below compare the many different device designs for D-mode high-power (Table 2), E-mode high-power (Table 3), and D-/E-mode RF applications (Table 4). Table 4 also includes the RF performance of mature GaN HEMTs and emerging hydrogen-terminated diamond HEMTs to illustrate the differences in performance of other material systems to β-Ga_2_O_3_.

### 3.1. β-Ga_2_O_3_ FET Structures

#### 3.1.1. MESFETs and Delta Doping

The metal–semiconductor field-effect transistor (MESFET) in Figure 3a, fabricated by Higashiwaki et al., was the first demonstrated single-crystal β-Ga_2_O_3_ transistor [101]. Many of the following MESFETs, reported by the Rajan group, incorporated delta doping [60,88,102,103,104,105,106]. The delta-doping technique was first developed in 2017 by Krishnamoorthy et al. [102] in attempts to improve Si doping during PAMBE epi layer growth. The Si source oxidized quickly, reducing the Si doping level in the β-Ga_2_O_3_, creating doping spikes. Pulsing the Si shutter for 1 s over a period of 1 min removed the oxide and created uniform, highly doped regions with UID spacers (Figure 3b), leading to the delta-doping method for β-Ga_2_O_3_ devices. This resulted in a 2D electron gas (2DEG), high-electron-mobility transistor (HEMT) behavior with an increased carrier sheet concentration and mobility, as well as reduced contact and sheet resistances. These improvements and lower gate capacitance of MESFETs compared to MOSFETs lend delta-doped MESFETs more for RF applications. Regrown ohmic contacts, discussed in the Section “Regrown Layers”, are required for delta-doped FETs to reach the 2DEG because it is surrounded by UID β-Ga_2_O_3_. The Rajan group fabricated delta-doped MESFETs using regrown contacts, gate-connected field plates (GFP), and highly scaled T-gate structures with gate lengths (L_G_) down to 120 nm to improve their low- and high-frequency performance, with a BFOM of 118 MW cm^−2^ [60], a mobility of 95 cm^2^ V^−1^ s^−1^ [104], and a current gain cutoff frequency (f_T_) of 27 GHz (Figure 3c) [88]. The GFP and T-gate structures are also discussed in Section 3.1.5 and Section 3.1.7, respectively.

Bhattacharyya et al. reported high-performing, non-delta-doped lateral MESFETs using a combination of regrown ohmic contacts for a low contact resistivity of 8.3 × 10^−7^ Ω cm^2^ [107], GFP for a V_br_ up to 4.4 kV [108,109], and a fin-shaped channel design surrounded by variable-temperature MOCVD-grown layers, achieving a mobility of 184 cm^2^ V^−1^ s^−1^, negligible hysteresis, and a BFOM of 0.95 GW cm^−2^ [63]. Passivating layers such as Al_2_O_3_ and SiN_x_ can also be used to improve both low-frequency BFOMs and high-frequency Huang’s Material Figure of Merit (HMFOM) [63,66]. The current highest-reported FET breakdown voltage of 10 kV was achieved using a MESFET design with a T-gate structure, source-connected field plates (SFPs), SiN_x_ passivation, oxygen annealing (OA), Si ion implantation, UID buffer layers surrounding the channel region, and B implantation for device isolation [59]. These device improvements will be discussed in more detail in later sections.

#### 3.1.2. Self-Aligned Gate (SAG) FETs

The self-aligned gate (SAG) FET design is a well-known process developed to reduce series resistance and aggressively scale devices by reducing the source–gate spacing (L_SG_), essentially eliminating the source–gate access region. The earliest β-Ga_2_O_3_ SAG FETs, developed by AFRL, were designed by first depositing the Al_2_O_3_ gate dielectric using plasma-assisted atomic layer deposition (PA-ALD), which acts as the ion implantation cap. A refractory metal gate of W or W/Cr, able to withstand the high ion activation temperature, is patterned to protect the gate and drift regions. The source–gate and drain–gate access regions are then very highly doped via Si ion implantation and activated at 900 °C using rapid thermal annealing (RTA) for 2 min in N_2_ ambient conditions [89,110]. The gate metal is then etched via reactive ion etching (RIE) from the drift region and the ohmic contacts are formed (Figure 4a). A low contact resistance (R_C_) of 1.5 Ω mm, a sheet charge density (n_s_) of 4.96 × 10^12^ cm^−2^, and a Hall mobility of 48.4 cm^2^ V^−1^ s^−1^ were measured in these devices [110]. RF load–pull continuous wave (CW) power measurements were reported for these early SAG FETs, with a high output power (P_out_), transducer gain (G_T_), and power-added efficiency (PAE) of 715 mW mm^−1^, 13 dB, and 23.4%, respectively, at 1 GHz (Figure 4a) [89,111].

A recent PAMBE-grown delta-doped SAG FET structure incorporated delta doping, in situ Ga etching for gate recess, and in situ Al_2_O_3_ gate dielectric growth, achieving sub-100 nm source–gate and gate–drain access regions [61]. A 30 nm Mg-doped layer was initially grown to compensate the Si impurities at the substrate/epi interface, followed by a 500 nm UID buffer layer. Then, two delta-doping layers, 5 nm apart, another 40 nm UID layer, and a 45 nm n++ cap layer were grown. The n++ cap layers were used as an alternative to ion implantation, enabling SAG. The SAG fabrication process (Figure 4b) began with ohmic contact fabrication and the plasma-enhanced chemical vapor deposition (PECVD) of SiN_x_ with patterning to expose the gate region. The sample was placed into an MBE system, where in situ Ga etching of the n++ cap layer was performed at a substrate temperature of 550 °C and Ga flux of 1.5 × 10^−7^ Torr. Ga droplets were removed at a temperature of 600 °C, followed by 10 nm in situ Al_2_O_3_ deposition at a temperature of 400 °C. Conformal ex situ ALD was used to uniformly deposit 60 nm of Al_2_O_3_ in the gate and sidewall regions. Anisotropic RIE and isotropic BOE wet-etching of the Al_2_O_3_ reduced the gate and sidewall dielectric thickness to 20 nm and 50 nm, respectively. This FET outperformed the initial SAG FETs, with a source–gate access resistance of 1.3 Ω mm, an n_s_ of 2.8 × 10^13^ cm^−2^, and a mobility of 65 cm^2^ V^−1^ s^−1^, leading to record peak DC and pulsed drain currents of 560 mA mm^−1^ and 895 mA mm^−1^, respectively, for a lateral FET on a native β-Ga_2_O_3_ substrate. The FET exhibited a high gate leakage and low current on/off ratio due to a low-quality gate dielectric or remaining Ga droplets at the interface. A drain current droop was observed in the DC measurements, indicating excessive self-heating, as shown by the red dashed line in Figure 4b. The use of SAG FETs has not yet been translated to vertical devices but is expected to improve performance for both low- and high-frequency operation.

#### 3.1.3. Trench/Recessed-Gate FETs

Another FET design, first implemented by AFRL in 2017 [90], is the trench or recessed gate to scale devices down to sub-µm gate lengths for improved RF performance. The FET in ref. [90] was fabricated on an n+ 180 nm channel layer with an n++ 25 nm ohmic cap layer grown via MOVPE. Post-ohmic contact formation, the n++ cap layer was etched and 200 nm of SiO_2_ was deposited via PECVD as the passivation and field-plate dielectric. A 0.7 µm gate region was patterned on the SiO_2_ and etched via RIE nearly halfway into the epi layer, followed by ALD-Al_2_O_3_ deposition as the gate dielectric, Ni/Au gate stack evaporation, and the evaporation of interconnects (Figure 5a). The measured cutoff frequency (f_T_) and maximum oscillating frequency (f_MAX_) were 3.3 GHz and 12.9 GHz [90].

E-mode FETs are useful in reducing off-state power loss; however, they are difficult to fabricate in β-Ga_2_O_3_ due to the lack of p-type doping, large hole effective mass, and hole self-trapping. The recessed-gate approach is among the few early methods used to achieve E-mode operation by etching into the channel region, such that the remaining channel is fully depleted due to band bending at the oxide/epi and epi/substrate interfaces [112,113]. Chabak et al. [80] studied the band bending of a 200 nm Si-doped 5.5 × 10^17^ cm^−3^ epi due to 5.5 × 10^12^ cm^−2^ surface states at the SiO_2_/β-Ga_2_O_3_ interface, noting an approximately 100 nm depletion, and 34 nm of depletion due to the Fe-doped substrate. The E-mode FET with a threshold voltage (V_th_) of +2 V was realized due to a gate recess of 140 nm (Figure 5b). E-mode recessed-gate FETs with an epi thickness of 200 nm, etch depth of 180 nm, and an L_G_ of 2 µm were also reported to have high switching characteristics, with turn-on/off delay times of 4.0 ns/11.8 ns and rise/fall times of 24.6 ns/82.2 ns (Figure 5c). The longer fall time is attributed to low electron mobility and slow discharging from interface states. While switching losses are reduced with higher switching speeds, a high on-resistance (R_on_), identified based on the V_DS_ of ≈5 V in the top plot of Figure 5c, results in high on-state power losses that might be more limiting than switching losses [114]. A high R_on_ and increased power losses are observed for many trench FETs; however, further L_G_ scaling by incorporating SAG can reduce the channel resistance contribution.

**Figure 5 materials-16-07693-f005:**
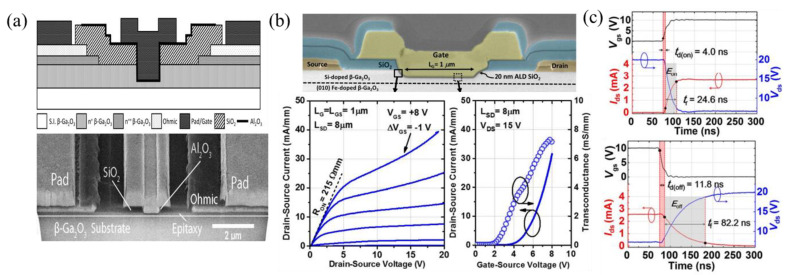
(**a**) First recessed-gate FET reported with sub-µm L_G_. Reprinted with permission from [90]. (**b**) First E-mode recessed-gate FET and respective transfer and output curves. © (2018) IEEE. Reprinted with permission from [80]. (**c**) Switching characteristics of a recessed-gate lateral FET where turn-on delay time, t_d(on)_, is defined as the time between 0.1V_gs_ and 0.1I_ds_. Likewise, t_d(off)_ is the time between 0.9V_gs_ and 0.9I_ds_. Similarly, the rise time, t_r_, is the time between 0.1I_ds_ and 0.9I_ds_, and the fall time, t_f_, is the time between 0.9I_ds_ and 0.1I_ds_. © (2019) IEEE. Reprinted with permission from [114].

Various TCAD studies have been reported on the effects of doping and recess depth in E-mode recessed-gate FETs. The band diagram variation, electron concentration, and V_th_ at different channel thicknesses (epi thickness–recess depth) is shown in Figure 6a. At zero gate and drain bias, the electron concentration drops off very quickly as the channel thickness drops below 80 nm, down to ≈10^8^ cm^−3^ at 50 nm, because of the oxide/epi and substrate/epi depletion. A threshold voltage near 0 V is observed for a 75 nm channel thickness going up to +4 V for 50 nm [115]. Decreasing the doping concentration in the channel layer is observed to both decrease the peak drain current and increase the V_th_, such that a doping of 1 × 10^16^ cm^−3^ causes E-mode behavior (Figure 6b). A larger recess depth (smaller channel thickness) increased the V_th_ from ≈−50 V to near 0 V from the I-V transfer curves but decreased the drain current in the I-V output curves (Figure 6b) [116]. At high V_GS_ and V_DS_, however, the drain current is nearly equal, indicating that the recess depth has little actual effect on the peak drain current. A slightly different trench FET design using body and epitaxial drift layers with different dopings, as well as recess through the entire drift layer, is shown in Figure 6c [117]. From I-V transfer curves with a drift layer doping of 3 × 10^17^ cm^−3^ and varying body doping from 1 × 10^13^ to 1 × 10^17^ cm^−3^, E-mode operation was only realizable for a body doping of 1 × 10^15^ cm^−3^ or less, showing a larger current and more negative V_th_ for higher doping concentrations. A 2D view of the electron concentration at a V_GS_ of 0 V, N_Body_ of 1 × 10^15^ cm^−3^, and N_Drift_ of 3 × 10^17^ cm^−3^ (Figure 6c) shows normally off conditions due to the full depletion of the body layer from band bending at the oxide/body interface.

#### 3.1.4. FinFETs

The first FinFET structures were lateral devices designed by Chabak et al. in 2016 using inductively coupled plasma (ICP) over-etching into the substrate to create thin 300 nm triangular-shaped fins as the channel (Figure 7a) [113]. E-mode operation was enabled by channel depletion due to the gate, with the I-V transfer curve shown in Figure 7a. Substrate conduction, shown by the red curve in Figure 7a, was observed and attributed to uncompensated carriers at the substrate surface. Hu et al. fabricated various vertical single-fin E-mode FETs [118,119,120] with current densities reaching 1 kA cm^−2^, a V_br_ of 1.6 kV, and a low subthreshold slope (SS) of 80 mV dec^−1^, giving an interface trap state density (D_it_) of >6 × 10^11^ cm^−2^ eV^−1^. Interface traps were observed to reduce the field-effect mobility and current density by depleting the channel, as well as limiting breakdown by exacerbating drain-induced barrier lowering (DIBL) [118]. Single- and multi-fin E-mode FETs were later fabricated by Li et al. with single/multi-fin current densities of 2 kA cm^−2^/230 A cm^−2^, a R_on,sp_ of 35.2 mΩ cm^2^/25.2 mΩ cm^2^, and BFOMs of 172 MW cm^−2^/280 MW cm^−2^ for a fin width (W_fin_) of 0.15 µm. Another advantage of multi-fin FETs is that, unlike single-fin FETs, current spreading does not drastically change the active area, making the BFOM and R_on,sp_ less ambiguous. The fabrication was performed on a 10 µm HVPE epi layer with doping of 2 × 10^15^ cm^−3^ grown on a conductive substrate. First, the epi layer was Si-ion implanted and activated at 1000 °C for the source ohmic contact, followed by e-beam lithography and dry etching to form sub-µm fin channels. A Ti/Au stack was deposited on the backside as the drain contact and a 35 nm ALD-Al_2_O_3_ was used as the gate dielectric. A sputtered Cr gate metal and 120 nm ALD-Al_2_O_3_ spacer were patterned with an SAG process. Finally, a Ti/Al/Pt stack was sputtered, forming the source and source-connected field plate. The devices were measured before and after post-deposition annealing (PDA) at 350 °C in N_2_, resulting in significant improvements (Figure 7b) [81]. It has been shown that V_th_ strongly decreases with increasing W_fin_ and N_D_ (Figure 7c), giving a small window for normally off devices. The previously mentioned FinFETs were fabricated with sub-0.5 µm W_fin_ and N_D_ below 1 × 10^16^ cm^−3^, enabling a positive V_th_. Because Si doping below 3 × 10^15^ is difficult in epi growth, a resistive layer via nitrogen doping of 1 × 10^16^ cm^−3^ during HVPE growth was shown to significantly minimize the V_th_ dependence on W_fin_ and achieve normally off operation for W_fin_ up to 2 µm (Figure 7c) [121].

Other vertical FinFETs have been reported on (100) oriented substrates to potentially reduce intrinsic growth defects, even though the majority are fabricated on (001) substrates [122]. Lateral tri-gate FinFETs have also reported high RF performance [91], high BFOMs of 0.95 GW cm^−2^, and mobilities of 184 cm^2^ V^−1^ s^−1^ using high/low temperature MOCVD growths [63], as mentioned in Section 3.1.1.

A highly selective wet-etching technique, metal-assisted chemical etching (MacEtch), is an attractive, damage-free alternative to dry etching that is typically used in FinFET fabrication [123]. More details on the MacEtch techniques and chemical reactions can be found in Ref. [124]. Lateral FinFETs fabricated via MacEtch have recently been reported (Figure 8a), with an aspect ratio of 4.2:1, an R_on,sp_ of 6.5 mΩ cm^2^, and a BFOM of 21 MW cm^−2^ [125]. The lowest SS, V_th_, and hysteresis of 87.2 mV dec^−1^, −6.9 V, and 24 mV, respectively, were measured on FinFETs with a 90° orientation from the [102] direction (Figure 8a). A previous study showed that fins perpendicular to [102] had the most vertical sidewalls and lowest D_it_ of 2.73 × 10^11^ cm^−2^ eV^−1^ [126]. DC I-V measurements of these FinFETs at high temperatures up to 298 °C (Figure 8b) reported increasing off-state currents and a lower on/off ratio, attributed to thermionic emission from source to drain, a decrease in V_th_ by ≈20 V due to trapping/de-trapping at the gate metal/oxide and oxide/semiconductor interfaces, and increasing hysteresis up to 4.29 V and SS up to 1.35 V dec^−1^, indicating thermal degradation of the interface or dielectric [127].

#### 3.1.5. Gate-Connected Field Plates

It is widely known that field plates can improve device breakdown by reducing the peak electric field near the contact edges. The gate-connected field plate (GFP) extends into the gate–drain access region, where most of the voltage drop occurs, “spreading” the electric field. The first β-Ga_2_O_3_ GFP FET was reported by Wong et al., using SiO_2_ as the FP dielectric (Figure 9a). TCAD simulations of the peak electric field for various field-plate heights, h_FP_, and field-plate drain lengths, L_FP,D_, at the drain edge of the gate (top), depicted by the symbol x in the diagram, and drain edge of FP (bottom), depicted by the symbol * in the diagram, are shown in Figure 9a. Increasing the L_FP,D_ is seen to quickly reduce the electric field at the gate edge, while having little effect on the field at the FP edge. However, as the h_FP_ increases, the field at the gate edge rises while the field at the FP edge falls, indicating an ideal window for h_FP_ [128].

Since then, other SiO_2_ FPs, SiO_2_ composite FPs with polymer passivation, and SiN_x_ FPs passivated with SiO_2_ have been reported [64,92,108,109,129,130,131] with some of the highest breakdown voltages and BFOMs of 8.56 kV and 355 MW cm^−2^, respectively. SiN_x_ is better suited both to spread electric fields due to its higher dielectric constant and in mitigating virtual gate effects originally discovered in AlGaN/GaN HEMTs [132], but is also mentioned as a possible mechanism of current dispersion [133] and series resistance increases [134] in β-Ga_2_O_3_ FETs.

**Figure 9 materials-16-07693-f009:**
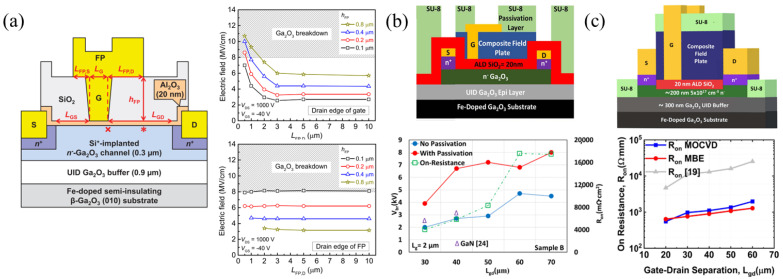
(**a**) GFP FET cross-section with the symbols x and * indicating peak electric fields in the channel. Plots of simulated breakdown electric field dependence on L_FP,D_ and h_FP_ are shown at the location of symbol x (top plot) and * (bottom plot). © (2016) IEEE. Reprinted with permission from [128]. (**b**) FET with composite PECVD-SiO_2_/ALD-SiO_2_ GFP and SU8 passivation used to increase V_br_. © (2020) IEEE. Reprinted with permission from [131]. (**c**) GFP FET similar to that in (**b**) but with SU8 as part of the FP and vacuum annealing, increasing V_br_ and reducing R_on_. © (2022) IEEE. Reprinted with permission from [64].

Zeng et al. in the Singisetti group used a composite FP composed of a thick 350 nm PECVD-SiO_2_ below a denser, high-quality 50 nm ALD-SiO_2_ layer to improve breakdown [129,130]. Sharma et al., also in the Singisetti group, then improved on the GFP design by adding a polymer, SU8, passivation layer on the composite FP and S/D regions, reaching some of the highest reported breakdown voltages of 8.03 kV (Figure 9b) and 8.56 kV (Figure 9c) [64,131]. The high R_on_ led to low BFOMs, but vacuum annealing before FP deposition resulted in a ×10 reduction in R_on_, with little change in V_br_ (Figure 9c) [64].

#### 3.1.6. Source-Connected Field Plates

Source-connected field plates (SFPs) are another viable FP strategy where the source metal extends past the gate, and can be considered a better field-spreading method across the gate region and at the drain side of the gate [135,136]. One of the first SFP FETs measured a BFOM of 50.4 MW cm^−2^ in 2019 (Figure 10a) [137]. Simulated electric field profiles in Figure 10a show field spreading and a reduced peak electric field using an SFP. A T-gate structure can be used in conjunction with SFPs for further field management, achieving higher BFOMs of 277 MW cm^−2^ (Figure 10b [65]) and a record V_br_ of 10 kV (Figure 10c [59]).

#### 3.1.7. T-Gates

T-gates, as mentioned previously, are unique in that they not only improve breakdown as a field-plate structure [59,62], but also improve RF results in thin-channel FETs by decreasing the L_G_ while maintaining a large cross-section. This reduces the gate access resistance and decreases the electron transport time, but does not degrade the noise figure [138]. Various T-gate RF FET structures are shown in Figure 11a–e that incorporate recessed gates with SiO_2_ FP dielectrics (Figure 11a [139]), air FP dielectrics with implanted channels (Figure 11b,c [94,140]), MESFET with Al_2_O_3_ passivation (Figure 11d [66]), and SiO_2_ gate dielectrics with SiN_x_ passivation (Figure 11e [93]). The FET in Figure 11e has the highest-reported f_max_ to date of 48 GHz and a high breakdown field of 5.4 MV cm^−1^. A T-gate MESFET with an f_T_ of 27 GHz was also discussed and shown in Figure 3b [88]. RF FETs with a T-gate structure must use highly scaled L_G_, typically in the range of 100–300 nm, for peak RF performance.

#### 3.1.8. Semiconductor-on-Insulator (SOI)

Another important property of β-Ga_2_O_3_ is the anisotropic cleavage planes, making the (100) plane easy to exfoliate as a nanomembrane, similar to graphene. This makes the fabrication of β-Ga_2_O_3_ devices on different substrates, or heterojunctions with unconventional materials such as transition metal dichalcogenides (TMDs), much simpler. The first SOI β-Ga_2_O_3_ FET was reported in 2014, with an exfoliated β-Ga_2_O_3_ layer placed on a p+ Si wafer with 285 nm of thermally grown SiO_2_ acting as the gate oxide [141]. The SOI FET was then fabricated via back-gate metal and top-source/drain ohmic contact deposition. The corresponding I-V curves proved that channels could be created using the mechanical exfoliation of β-Ga_2_O_3_. Other p+ back-gate SOI FETs have been fabricated and studied [68,82,142,143,144,145,146,147,148,149,150,151]. One advantage of SOI FETs compared to non-SOI FETs is that more devices can be fabricated from β-Ga_2_O_3_ wafers and, therefore, studies on transport, irradiation, thermal effects, etc., can be more cheaply and readily performed. Different methods of tuning the V_th_ have been implemented, such as by varying the β-Ga_2_O_3_ channel layer thickness [82,145], fluorine plasma [147], adding a p-type material such as p-SnO on the channel of back-gate FETs [85], and using a fixed back-gate bias (V_BG_) on top-gate FETs. A top-gate FET with V_BG_ shows variation in transconductance (g_m_) and V_th_ with V_BG_, obtaining a V_th_ of 0 V for V_BG_ ≈ 6 V (Figure 12a) [152]. Other studies about defect impacts on current dispersion [148], proton irradiation [144], scattering mechanisms [153], and device improvement using thermal management via different thermally conductive substrates have been reported using SOI FETs [71,73,154,155,156,157,158,159]. These will be discussed more in the Sections “AlN/GO”, “SiC/GO”, and “Diamond/GO”. High-performing SOI FETs have also been realized, with the highest reported mobility of 191 cm^2^ V^−1^ s^−1^ (Figure 12b [85]), high current densities reaching 1.5 A mm^−1^ (Figure 12c [68]), an SS of 61 mV dec^−1^ very near the thermionic limit using a TMD-TaS_2_/β-Ga_2_O_3_ heterojunction (Figure 12d [160]), a V_br_ up to 800 V [67], and BFOMs of 100 MW cm^−2^. The high BFOM of 100 MW cm^−2^ is achieved by a β-Ga_2_O_3_-on-SiC FET using ion cutting, a novel heterogenous wafer-scale integration technique for β-Ga_2_O_3_ on SiC [156,159].

Other SOI FETs have integrated β-Ga_2_O_3_ nanomembranes with various p-type 2D materials such as WSe_2_ [161,162], MoTe_2_ [162], and black phosphorus (BP) [163] to act as p-type gates, and large work function materials such as NbS_2_ and TaS_2_ to improve SS (61 mV dec^−1^) and off-state behavior [160]. Double-gate FETs using top-gate dielectrics such as HfO_2_ [152], h-BN [164], and bottom-gate dielectrics as the SiO_2_ on p-Si wafers have been utilized for improved gate control and V_th_ tuning. Monolithically integrated top and bottom graphene gates with both E-mode and D-mode FETs on the same layer are one of the first mentions of a β-Ga_2_O_3_ logic circuit with both E-mode and D-mode FETs [165].

#### 3.1.9. Other Novel Structures

Most of the designs discussed in the previous sections were first developed in the early stages of β-Ga_2_O_3_ devices and iteratively improved upon. With the help of TCAD simulation, novel structures with high potential can be initially proposed. One of the recently proposed structures (2022–2023) includes a lateral field-plated MOSFET with a self-aligned trench vertical gate (Figure 13a [166]). The channel from source to drain starts with a UID β-Ga_2_O_3_ buffer layer with ion implantation in the source region. The UID buffer layer separates the ion-implanted source to the n+ horizontal channel, reaching the drain. The trench portion of the gate falls below the n+ channel into the UID buffer layer. The dominant channel becomes the vertical UID portion, which is highly controlled and not restricted by high-resolution photolithography. This structure has been proposed for AlGaN/GaN HEMTs and shown to improve drain current and transconductance [167]. The similar β-Ga_2_O_3_ device incorporates a GFP with a SiO_2_ FP dielectric to improve V_br_ and current uniformity in the channel.

Another proposed device is a lateral MOSFET with an SFP, where the FP dielectric is air/SiN_x_ and the FP makes contact with the SiN_x_ in the gate–drain drift region (Figure 13b [168]). The electric field plot (Figure 13b) mentions device 1 (the proposed device), device 2 incorporating a GFP, and device 3 with no FP. The proposed device exhibits a higher BFOM of ≈2.2 GW cm^−2^ compared to 1.6 GW cm^−2^ for device 2 and 106 MW cm^−2^ for device 3. The capacitances, C_gd_ and C_gs_, of the air gap device were slightly higher than the non-FP device, resulting in a slightly lower BHFOM, but an overall much larger JFOM of 7.8 THz V.

Gate-all-around (GAA) FETs are another newer FET used to scale below 10 nm in Si, but has not yet been realized in β-Ga_2_O_3_. A proposed β-(AlGa)_2_O_3_/Ga_2_O_3_ GAA FP HEMT is simulated with a high P_out_ of ≈22 kW and an f_T_ of 2.4 GHz, showing potential for future GAA β-Ga_2_O_3_ FETs (Figure 13c [169]).

The lack of p-doping and very low hole mobility in β-Ga_2_O_3_ has limited most devices to unipolar operation. However, recently, there has been a TCAD investigation of potential β-Ga_2_O_3_ heterojunction bipolar transistors (HBTs) using p-type oxides. An npn with p-CuO_2_ showed HBT behavior and current gain (Figure 13d), but both the current gain and the breakdown electric field are strongly limited by interface traps and the low bandgap of CuO_2_ (2.1 eV) [170]. They mention that other p-oxides like NiO could apply, and that an (Al_x_Ga_1−x_)_2_O_3_ layer could be used as the emitter to reduce the electron barrier into the base. For future designs, they propose specifications for minority carrier transport, emitter-base CBO, and a threshold for interface trap state density.

### 3.2. Channel and Substrate Materials

This section will mainly discuss β-Ga_2_O_3_ FETs designed by using different materials and processes as opposed to, in the previous section, FET structures and patterning.

#### 3.2.1. Current Aperture Vertical Transistors and U-Shaped Trench MOSFETs

Vertical FETs are more suited for high-power applications than lateral FETs due to their higher power densities and smaller size, because breakdown voltage scales with drift layer thickness, as opposed to lateral devices where it scales with L_GD_, sacrifice the chip area. Current aperture vertical transistors (CAVETs), motivated by their counterparts in Si [171], SiC [172], and GaN [173], use current-blocking layers (CBLs) to reduce off-state drain currents and improve on/off ratios. The CBLs can either surround the source from the drift layer for E-mode purposes (Figure 14a [83]), or leave an opening/aperture for carriers to access the channel. For the later type, E-/D-modes are dependent on the doping of the channel [69,83,174,175,176]. CAVETs with only the channel doping, n_ch_, varying from 5 × 10^17^ to 1.5 × 10^18^ cm^−3^ show E-mode behavior for a doping of 5 × 10^17^ cm^−3^ and D-mode behavior for the other dopings (Figure 14b [174]). Varying the aperture length, L_ap_, creates diode-like behavior with increasing on-voltage, V_on_, as the L_ap_ decreases, possibly due to a fixed sheet charge of 10^11^–10^12^ cm^−2^ originating from defects diffused from the CBL (Figure 14c [176]). Semi-insulating CBLs using Mg^2+^ ion implantation were first used, but resulted in high leakage currents due to large Mg diffusion during the activation anneal [175]. The nitrogen thermal diffusivity in β-Ga_2_O_3_ is much lower compared to Mg [177], and therefore, CBL layers formed via N^2+^ ion implantation have exhibited less leakage [69,174,176,178].

The U-shaped trench-gate MOSFETs (UMOSFETs) are known to increase the packing density and reduce input capacitance when compared to planar vertical FETs. They have only recently been studied in the β-Ga_2_O_3_ material system using CBLs (Figure 14d), with both oxygen annealing, discussed in the following section, and N ion implantation [84,179]. Oxygen annealing formed an insulating CBL but increased the contact and sheet resistance in the n+ layers used for source ohmic contacts, while N implantation did not have these setbacks and showed higher current densities. One major drawback of these devices so far is the low on/off ratio of ≈7 × 10^4^.

#### 3.2.2. Oxygen Annealing

Oxygen annealing (OA), mentioned in the previous sections, increases the resistivity of n-type β-Ga_2_O_3_. The intended mechanism is reducing oxygen vacancies, which act as deep donors, thereby increasing acceptor compensation [180]. However, there is still some uncertainty in the true mechanism due to the multiple Ga and O sites in the unit cell, and complex substitutions that can occur at high temperatures [181,182]. OA was first observed to reduce the effect of a secondary conducting channel due to the UID layer, improving the pinch-off, output power density, and high-speed performance [95]. While there are few reports of OA FETs, there is potential for the future incorporation of OA (Figure 10c) [59,84,183].

#### 3.2.3. Heterostructures

##### (Al_x_Ga_1−x_)_2_O_3_/β-Ga_2_O_3_ Modulation-Doped FETs

The (Al_x_Ga_1−x_)_2_O_3_/β-Ga_2_O_3_ (AlGO/GO) heterostructure has been studied intensively because of the observed carrier confinement near the AlGO/GO interface due to the conduction band offset (CBO) of ≈0.6 eV [184,185]. The first modulation-doped FETs (MODFETs) used a Ge-doped AlGO layer between two UID-AlGO layers to generate a 2DEG below UID-β-Ga_2_O_3_ [186]. However, the majority of later MODFETs incorporated delta doping in the AlGO layer which, along with the AlGO/GO CBO, generated a 2DEG inside the UID-β-Ga_2_O_3_ near the AlGO/GO interface. This avoids reductions in mobility due to the dopants of the 2DEG (Figure 15a) [187,188]. Like delta-doped FETs, regrown ohmic contacts are typically used to reach the 2DEG.

Various enhancements—such as double-heterostructure MODFETs with a UID β-Ga_2_O_3_ quantum well (Figure 15b [189]) and reduced spacer lengths, defined as the distance between the β-Ga_2_O_3_ and delta doping, down to 1 nm [190,191]—are used to increase carrier concentration. The 2DEG charge density typically is in the range of 1 × 10^12^–5 × 10^12^ cm^−2^, while the highest of 1.1 × 10^13^ cm^−2^ was measured in a double heterostructure, 1 nm spacer, high-k gate dielectric MODFET [79]. Field plates [70] and high-k gate dielectrics have shown to improve breakdown up to 5.5 MV cm^−1^ [79].

A modification of the MODFET, named the heterostructure FET (HFET), uses a heavily doped AlGO spacer as opposed to a delta-doping layer with a UID spacer (Figure 15c). E-beam lithography scaled L_SG_ down to 55 nm, reducing the parasitic resistance, reporting near-record f_T_ and f_max_ values of 30 GHz and 37 GHz, respectively, and minimal RF degradation at temperatures up to 250 °C [96,97].

##### AlN/GO

AlN is potentially a better alternative to AlGO/GO MODFETs with its higher CBO to β-Ga_2_O_3_ of ≈1.7–1.86 eV and polarization-induced charge, leading to larger 2DEG concentrations of 3 × 10^13^–5 × 10^13^ cm^−2^ [192,193,194]. Currently, no AlN/β-Ga_2_O_3_ HEMTs have been fabricated and reported; however, multiple TCAD simulations show promise, with much higher frequency operations up to an f_T_ of 166 GHz and f_max_ of 142 GHz [195,196,197].

The thermal heat sink advantages of AlN, with a thermal conductivity of ≈320 W m^−1^ K^−1^ [25], to β-Ga_2_O_3_ has also been studied in SOI MOSFETs on an AlN/Si substrate for heat dissipation, showing little-to-no current dispersion between DC and pulsed I-V (Figure 16a [71]).

##### SiC/GO

SiC, with its high thermal conductivity of 370 W m^−1^ K^−1^, is mainly used for heat dissipation when forming a heterostructure with β-Ga_2_O_3_ [25]. TCAD simulations of p-SiC replacing the semi-insulating substrate in β-Ga_2_O_3_ FETs reduced peak temperatures by 100 °C. High breakdown voltages and on-currents were maintained by increasing the SiC thickness and the β-Ga_2_O_3_ doping to avoid premature breakdown in SiC [198,199]. Both the growth [200] and process/integration methods [156,201] have been developed for β-Ga_2_O_3_/SiC heterojunctions. Fabricated transistors have used the SiC layer only as a thermal heat sink (Figure 16b) [72,159], while others have used p-SiC to also behave as a back gate [158]. Figure 16b shows a composite SiC/β-Ga_2_O_3_ MOSFET formed via fusion bonding [201] with a large reduction in the temperature rise with a SiC substrate compared to a β-Ga_2_O_3_ substrate [72].

##### Diamond/GO

Diamond has one of the highest, if not the highest, thermal conductivity of any semiconductor at approximately 2290 W m^−1^ K^−1^. For this reason, diamond is commonly used in high-power applications for heat dissipation, such as cooling mechanisms with micro channels demonstrated in GaN [5]. TCAD simulations have confirmed the advantages of using a nanocrystalline diamond (NCD) substrate compared to β-Ga_2_O_3_ or SiC substrates [202]. In β-Ga_2_O_3_, thermal studies of both exfoliated nanomembranes on diamond substrates and the ALD growth of polycrystalline β-Ga_2_O_3_ on diamond substrates have been realized, with the nanomembrane reporting better thermal boundary conductance (TBC) due to the low thermal conductivity of polycrystalline β-Ga_2_O_3_ [157,203]. SOI MOSFETs on diamond exhibited higher drain currents of 980 mA mm^−1^ and a 60% reduction in temperature rise compared to similar devices on a sapphire substrate (Figure 16c) [73,154]. Additionally, a study on various device thermal-cooling methods concluded that the best performing solution in terms of the lowest temperature rise and thermal resistance consisted of cooling devices from the top contact nearest to the junction, named junction-side cooling, through flip-chip hetero-integration via thermal bumps to a diamond carrier, and NCD passivation with a thermal conductivity of 400 W m^−1^ K^−1^, labeled as FC3 in Figure 16d [204].

**Figure 16 materials-16-07693-f016:**
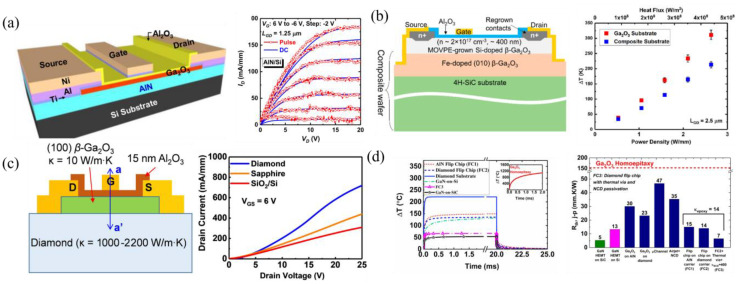
Thermal studies using AlN, SiC, and diamond. (**a**) SOI FET on an AlN/Si substrate showing effective heat dissipation when comparing DC and pulsed I-V. Reprinted from [71]. (**b**) SiC/GO composite wafer MOSFET with significant reduced temperatures at high power densities. Reprinted with permission from [72]. Copyright 2023 American Chemical Society. (**c**) MOSFET with diamond substrate and I-V curves compared to FETs, with other substrates showing significant current dispersion caused by self-heating. Reproduced from [73]. CC BY 4.0. (**d**) Simulation and comparison of device-level cooling methods. © (2019) IEEE. Reprinted with permission from [204].

### 3.3. Source and Drain Ohmic Contacts

This section covers the design of ohmic contacts, including metals, processes, and techniques, to decrease contact resistance and improve ohmic behavior.

#### 3.3.1. Metals and Processes

By far, the most common metal stacks for ohmic contact formation are Ti/Au or Ti/Al/Ni/Au due to the low metal work function of Ti (≈4.3 eV) [101,205]. In the earliest reports of β-Ga_2_O_3_ devices, BCl_3_ RIE was performed following metal evaporation and lift-off [206], but more recent reports use RTA at 400–500 °C for 1 min in N_2_. Comprehensive studies on the Ti/Au interfacial reactions found that the interdiffusion of Ti and Au, as well as a thin Ti-TiO_x_ interlayer partially lattice-matched to β-Ga_2_O_3_, is responsible for ohmic contact formation (Figure 17a) [207,208]. Additionally, Si ion implantation and RIE improved the thermal stability and lowered resistivity in Ti/Au ohmic contacts [209]. Ti/Au ohmic contacts were observed to perform better in the (001) and (−201) orientations compared to (010), potentially due to more dangling bonds and higher surface energy in the (001) and (−201) directions [210].

Yao et al. investigated the capability of forming ohmic contacts with various metals such as Ti, In, Ag, Sn, W, Mo, Sc, Zn, and Zr, with and without an Au capping layer, concluding that Ti/Au with RTA at 400 °C for 1 min in Ar resulted in ohmic behavior with the lowest contact resistance [211]. Temperatures at 500+ °C degraded the Ti/Au contact and increased the resistivity. In, with its work function of 4.1 eV, also showed ohmic behavior after annealing at 600 °C for 1 min in Ar, but is not practical due to its low melting point. All other metals exhibited pseudo-ohmic or non-ohmic behavior, concluding that Ti/Au is the ideal metal stack of the nine used.

Other metals that have been found to form ohmic contacts on β-Ga_2_O_3_ are Mg/Au, with a work function of 3.8 eV [212]. The electrode resistance was found to increase with increasing annealing temperature from 300 to 500 °C due to Mg oxidation. Annealing at 300 °C and 500 °C found that the current density was not consistent 37 days later, while that at 400 °C was the same, indicating long-term stability. In order to withstand high temperatures, a refractory metal alloy TiW ohmic contact is feasible to highly doped (1 × 10^19^ cm^−3^) β-Ga_2_O_3_ [213].

**Figure 17 materials-16-07693-f017:**
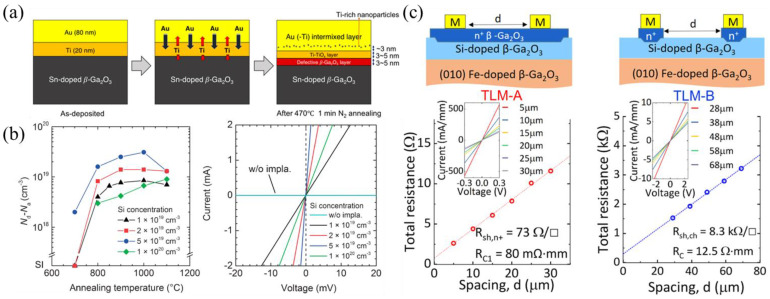
(**a**) Formation of a defective β-Ga_2_O_3_ and TiO_x_ layer enabling ohmic contacts. Reproduced from [207]; licensed under a Creative Commons Attribution (CC BY) license. (**b**) Effective doping vs. annealing temperatures of Si implantation and I-V curves for various Si concentrations after annealing at 950 °C. Reproduced from [214]. © The Japan Society of Applied Physics. Reproduced with the permission of IOP Publishing Ltd. All rights reserved. (**c**) Regrown ohmic contacts reducing contact resistance as shown through TLM measurements of the regrown layer (TLM-A) and to the channel layer (TLM-B). Reproduced from [107]. © The Japan Society of Applied Physics. Reproduced with the permission of IOP Publishing Ltd. All rights reserved.

#### 3.3.2. Improvements

Decreasing the ohmic contact resistance improves both low- and high-frequency performance and is thus a critical component of FET design.

##### Ion Implantation

Si ion implantation was the first technique, starting in 2013 [214,215], to reduce contact resistance, and has been used in many future FET designs to form n+ layers in the source/drain regions before metal evaporation. In β-Ga_2_O_3_, ion implantation is achieved by first depositing a thick cap layer and patterning to expose the implant regions. Secondly, the implant depth and dose must be determined iteratively, followed by ion activation. For Si implantation, ions are typically activated at 900–950 °C for 30 min in N_2_ [66,216]. Figure 17b shows the annealing temperature effects on Si ion implant concentrations ranging from 10^19^ to 10^20^ cm^−3^, with an ideal temperature window in the range of 900–1000 °C. The I-V curves show the least resistance at a Si concentration of 5 × 10^19^ cm^−3^ for annealing at 950 °C [214]. Some use various implant steps for either more conductive regions or for semi-insulating dopants, like N or Mg, such as in CBL formation or some E-mode FETs [112,128,174,175]. Ge ion implantation for n+ regions has recently been investigated with pulsed RTA from 900 to 1200 °C for dopant activation. While one of the lowest specific contact resistivity (ρ_C_) values of 4.8 × 10^−7^ Ω cm^2^ was reported at 1100 °C, a strong redistribution of the ions was observed along with an increased surface roughness [217].

##### Regrown Layers

Regrown layers are an alternative to ion implantation that avoid damage caused by the high-energy ions and high annealing temperatures. The fabrication process for regrown layers is outlined as follows: a sacrificial layer (typically SiO_2_) is patterned, followed by etching through the SiO_2_ and a portion of the epitaxial layer in the exposed source/drain regions. The sample is then placed in a growth chamber, and an n+ layer for ohmic contact evaporation is grown, followed by wet-etching the sacrificial layer to lift-off the regrown layers outside the source/drain regions [104,107,108,109,187,189,218]. Besides avoiding the damage caused by ion implantation, regrown layers are commonly used to contact a 2DEG in a delta-doped FET or MODFET. Initial Si delta doping before regrowth is used to neutralize any F^−^ ions, incorporated during dry etching, that could deplete the channel, confirmed to have low contact resistance using the transfer length method (TLM) (Figure 17c [107]). Recently, ultra-low ρ_C_ values for both β-Ga_2_O_3_ and β-(Al_x_Ga_1−x_)_2_O_3_ of 1.62 × 10^−7^ Ω cm^2^ and 5.86 × 10^−6^ Ω cm^2^, respectively, have been reported, using dopings up to 3.2 × 10^20^ cm^−3^ [219].

##### Intermediate Layers

In WBG semiconductors, adding intermediate layers with lower bandgaps and/or higher doping concentrations can reduce the barrier for carrier transport to and from the contact. In β-Ga_2_O_3_, the most common intermediate layers are indium–tin–oxide (ITO) and aluminum–zinc–oxide (AZO). Both ITO and AZO, deposited via sputtering, have been used to form ohmic contacts. The ohmic contacts were formed on a 2 × 10^17^–3 × 10^17^ cm^−3^ doped β-Ga_2_O_3_ epi with varying annealing temperatures of 900–1150 °C and 500–600 °C for ITO [220,221], and 400–600 °C for AZO [222].

##### Diffusion Doping (Spin-on-Glass)

Diffusion doping, or spin-on-glass (SOG) doping, is one of the least common ways to both dope β-Ga_2_O_3_ and improve ohmic contact resistance. It provides a lower-cost and simpler fabrication process than the typical ion implantation or regrowth methods, as well as more predictable diffusion during activation and peak doping at the surface, ideal for ohmic contacts. The process begins with Sn-doped SOG spin-coated on a β-Ga_2_O_3_ epi layer and RTA at 1200 °C for 3–5 min in N_2_ to activate the dopants, followed by the removal of the SOG layer via dipping in buffered HF (BFH) for 10 min. A low ρ_C_ of 2.1 × 10^−5^ Ω cm^2^ was reported, and lateral MOSFETs reported improved peak current density and transconductance as well as a high thermal stability up to 200 °C using SOG [223,224].

### 3.4. Gate Dielectrics

#### 3.4.1. Materials and Processes

The choice of gate dielectric material is vital to high-performing β-Ga_2_O_3_ FETs. The main material used is (PA)-ALD Al_2_O_3_ because of its large bandgap of 6.4–6.9 eV, with both electron- and hole-blocking capabilities, and similar composition to β-Ga_2_O_3_ [225]. The first MOSFETs, as well as many of the MOSFETs discussed previously, use Al_2_O_3_ as their gate dielectric [205,215]. While the main deposition method is (PA)-ALD, some initial reports of using in situ MOCVD-grown Al_2_O_3_ immediately after β-Ga_2_O_3_ epi growth have garnered attention, measuring lower interface defect densities and higher-quality Al_2_O_3_ to improve breakdown characteristics [226,227].

The second most common gate dielectric is SiO_2_, with the advantage of a higher bandgap around 9 eV but a lower dielectric constant, which becomes important in distributing electric field profiles, discussed later. The deposition of SiO_2_ can be performed either via PECVD or ALD [129,130].

Low defect densities have been reported for Al_2_O_3_ using solvent, O_2_ plasma, piranha, and BHF surface cleaning before ALD deposition and in situ forming-gas post-deposition annealing (PDA) at 250 °C [228], as well as SiO_2_ with a solvent only [229] or both solvent and piranha cleaning [230].

#### 3.4.2. p-Gates

The more recently investigated gate dielectrics are p-type materials. The main materials used are p-NiO, p-GaN, and p-SnO. P-gated FETs, referred to as heterojunction FETs (HJ-FETs), are unique in that they provide vertical channel depletion for pinch-off while also increasing V_br_ due to the pn junction between the gate and channel. This allows for thicker and higher doped channel regions, leading to higher currents and lower R_on_, without a reduction in V_br_.

#### p-NiO

P-NiO has gained significant attention as a candidate for pn heterojunctions, with a wide bandgap of 3.7–4.0 eV and controllable p-doping in the range of 10^16^–10^19^ cm^−3^. Additionally, the highest-recorded BFOM of 13.2 GW cm^−2^ in β-Ga_2_O_3_ was recently reported on p-NiO/n-β-Ga_2_O_3_ heterojunction diodes [231]. P-NiO is typically sputtered at room temperature with a hole concentration modulated using the Ar/O_2_ ratio [75,232]. The theoretical CBO and VBO of p-NiO to β-Ga_2_O_3_ is expected to be 2.2 eV and 3.3 eV, respectively, while experimentally observed values vary greatly due to the polycrystalline p-NiO from sputtering [74,233]. P-NiO gated FETs have reported BFOMs of 0.39 GW cm^−2^ [74], and with incorporated recessed-gate p-NiO bi-layers, a T-gate structure with a SiO_2_ FP, and piranha treatment, a negligible hysteresis of 4 mV, an SS of 66 mV dec^−1^, and a BFOM of 0.74 GW cm^−2^ were realized (Figure 18a [75]). The recessed gate is also useful in creating E-mode FETs [234]. One difficulty, however, is the increased gate leakage which occurs when the pn junction becomes forward-biased. A proposed solution, studied initially through TCAD simulations and verified using fabricated devices, is the addition of a dielectric layer between the p-NiO and gate metal to suppress gate leakage when the pn junction is forward-biased [76,235]. A 20 nm ALD-SiO_2_ interlayer reduced the gate leakage by six orders of magnitude, maintained an on/off ratio of 10^6^, and improved the gate swing from 3 V to 13 V compared to the non-interlayer FET (Figure 18b).

As an aside, FETs using reduced surface electric field (RESURF) and superjunction (SJ) techniques incorporating p-NiO have shown to increase breakdown, although more work is required to improve their BFOMs [236,237].

#### p-GaN

P-GaN is another material that can be used as a p-gate to β-Ga_2_O_3_; however, it has only been studied in TCAD simulations. Similar to p-NiO, it is primarily to improve R_on_ without sacrificing V_br_, as well as to enable normally off operation by depleting the channel underneath. Increasing the doping and/or thickness of the p-GaN layer increases the V_th_, as more charge is available to deplete the channel [194,238]. Increased p-GaN doping is shown to have little effect on g_m_, while an increase in p-GaN thickness reduces g_m_ due to reduced gate control. In a vertical FinFET, the addition of p-GaN as the gate metal also improves V_br_ when compared to Schottky gate metals such as Ni/Au because of the higher work function of GaN, leading to a vertical electron barrier of ≈5 eV for p-GaN gates compared to ≈2.5 eV for a Ni/Au gate metal (Figure 18c). Additionally, the V_br_ of the p-GaN-gate FinFET is more resistant to W_fin_ increases compared to the Ni/Au-gate FinFET [239].

**Figure 18 materials-16-07693-f018:**
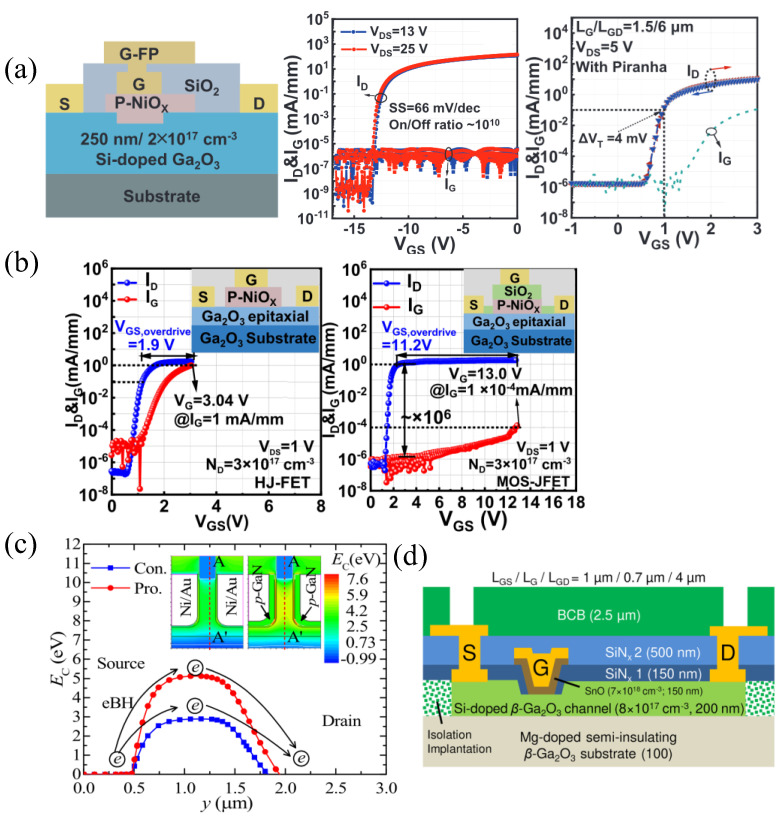
(**a**) P-NiO-gated HJ-FET with BFOM of 0.74 GW cm^−2^, ultra-low SS, and negligible hysteresis due to piranha treatment. Reproduced from [75], with the permission of AIP Publishing. (**b**) Addition of SiO_2_ between gate metal and p-NiO increases pn turn-on and enables larger gate swing. © (2023) IEEE. Reprinted with permission from [76]. (**c**) E_C_ electron barrier of FinFET with conventional (Con., blue) Ni/Au gate contact compared with proposed (pro., red) device using p-GaN as the gate metal, simulated through TCAD. Reproduced from [239]. © IOP Publishing. Reproduced with permission. All rights reserved. (**d**) HJ-FET with p-SnO-gate dielectric grown via PAMBE. Reproduced from [77]. CC BY 4.0.

#### p-SnO

MBE-grown p-SnO is reported to have a low bandgap of 0.7 eV and a type-I band alignment to β-Ga_2_O_3_, with CBO and VBO values of 0.49 eV and 3.70 eV, respectively. However, HJ-FETs have reported breakdown electric fields of ≈2 MV cm^−1^, possibly due to the high-quality MBE-grown SnO (Figure 18d) [77,240]. Another report of p-SnO was in an SOI FET using sputtered p-SnO as a back depletion layer, but not as a gate. This shifted the V_th_ to +40 V, and resulted in record high mobilities of 191 cm^2^ V^−1^ s^−1^ (Figure 12b [85]).

#### 3.4.3. High-k Gate Dielectrics

High-k dielectrics are a heavily studied field in β-Ga_2_O_3_ devices, particularly for their incredible field-spreading capability [241]. HfO_2_, BaTiO_3_ (BTO), and SrTiO_3_ (STO) are the most used high-k dielectrics, where BTO and STO are considered extreme-k dielectrics because their dielectric constants can be on the order of 300. Early reports of high-k FETs used ALD-HfO_2_, but both FETs suffered from high interface trap densities [86,242]. A later report of SOI FETs with ALD-HfO_2_ gate dielectrics reported near-ideal behavior, with negligible hysteresis, an SS of 64 mV dec^−1^, and an on/off ratio of 10^8^. This result was attributed to the high-temperature HfO_2_ deposition of 350 °C, forming a high-quality polycrystalline layer (Figure 12a [152]).

Xia et al. discuss the ability of extreme-k dielectrics in reducing the peak electric field and premature breakdown due tunneling by increasing the barrier length in heterojunction diodes [243]. Kalarickal et al. developed an electrostatic model of extreme-k dielectrics in FETs using the polarization of the charge to create highly uniform electric field profiles in the channel and enable both a high sheet charge density and an average breakdown field (Figure 19a). The modeled FET was then fabricated to show the improvement in the average breakdown field up to 4 MV cm^−1^ and a high sheet charge density of 1.6 × 10^13^ cm^−2^ using extreme-k dielectric BTO with a dielectric constant of 235 [78]. Kalarickal et al. improved on their FET design by first adding a 12.5 nm low-k dielectric of ALD-Al_2_O_3_ on the epitaxial layer to reduce interface traps and protect the surface during the sputtering of extreme-k BTO, resulting in an average breakdown field of 5.5 MV cm^−1^ and a BFOM of 408 MW cm^−2^ (Figure 19b [79]).

#### 3.4.4. Multi-Stack Gate Dielectrics

As previously mentioned, multi-stack gates can provide the advantages of both materials, such as in ref. [79], with a low-k Al_2_O_3_/extreme-k BTO multi-stack gate dielectric. Comparisons between the Al_2_O_3_/HfO_2_ and HfO_2_/Al_2_O_3_ gate stacks indicates reduced gate leakage in the HfO_2_/Al_2_O_3_/β-Ga_2_O_3_ because of the better carrier-blocking capability of the Al_2_O_3_. Both stacks show increased dielectric constants and similar D_it_ [244]. Another study comparing polycrystalline (p-)/amorphous (a-) HfO_2_ with p-HfO_2_/a-Al_2_O_3_ gate stacks showed that the p-HfO_2_/a-HfO_2_ stack had a lower D_it_, as evidenced by a distorted energy band in p-HfO_2_/a-Al_2_O_3_, a larger effective barrier height of 1.62 eV, and a hard breakdown field of 9.1 MV cm^−1^ compared to 4.9 MV cm^−1^ of the p-HfO_2_/a-Al_2_O_3_ stack (Figure 19c). The HfO_2_ bilayer stack also outperforms the single-layer p-HfO_2_ in breakdown because of better leakage suppression [245]. Similarly, the higher bandgap of SiO_2_ reduced gate leakage by 800× and increased the breakdown electric field by 1.7× when added as an interlayer between Al_2_O_3_/β-Ga_2_O_3_ [230]. Multi-gate stacks have also been used for ferroelectric charge storage via Al_2_O_3_/Hf_0_._5_Zr_0_._5_O_2_ (Al_2_O_3_/HZO) and Al_2_O_3_/HfO_2_/Al_2_O_3_/HZO, where the HZO polarization traps charge at the Al_2_O_3_/HZO interface for the first stack and in the HfO_2_ in the second stack (Figure 19d) [87,246].

**Figure 19 materials-16-07693-f019:**
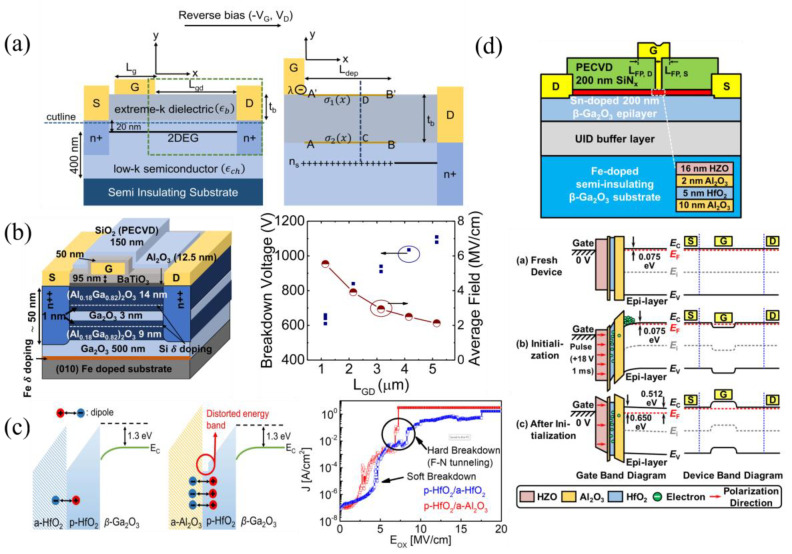
(**a**) The charge profile after applying a negative gate bias on a 2DEG with an extreme-k BTO gate dielectric. © (2021) IEEE. Reprinted with permission from [78]. (**b**) Double-heterostructure AlGO/GO MODFET with low-k Al_2_O_3_/extreme-k BTO gate dielectric reaching average breakdown electric fields of 5.5 MV cm^−1^. © (2021) IEEE. Reprinted with permission from [79]. (**c**) Multi-stack gate dielectrics of p-HfO_2_/a-HfO_2_ with superior breakdown compared to the p-HfO_2_/a-Al_2_O_3_ stack. © (2021) IEEE. Reprinted with permission from [245]. (**d**) Ferroelectric charge storage due to multi-stack gates and polarization trapping by Hf_0_._5_Zr_0_._5_O_2_. Reproduced from [87], with the permission of AIP Publishing.

## 4. Defect Engineering

### 4.1. Defects

Defects can severely degrade device performance and reliability and are thus an important area of study for any semiconductor device. WBG and UWBG semiconductors such as β-Ga_2_O_3_ require somewhat different characterization methods to study deep traps, since room-temperature and high-temperature measurements still only observe a small portion of the bandgap, and therefore photon excitation is typically used to obtain trap information throughout the bandgap. The following discusses many of the various trap characterization methods for β-Ga_2_O_3_, as well as the material preparations used to reduce traps in β-Ga_2_O_3_.

#### 4.1.1. Characterization

Deep-level transient spectroscopy (DLTS) and deep-level optical spectroscopy (DLOS) are powerful techniques to determine the energies and concentrations of deep-level traps. DLTS and DLOS are based on the principle that the capture and emission of carriers from traps in the space charge region (SCR) varies the measured capacitance. Therefore, capacitance transients are typically used to determine trap energy levels and their capture and emission rates at a specific energy level, determined by the temperature in DLTS and photon energy in DLOS. DLOS is required for (U)WBG semiconductors because most DLTS systems are limited to ≈1 eV below E_C_ or above E_V_, which is insufficient to fully characterize (U)WBG materials. The traps found in β-Ga_2_O_3_ via DLTS and DLOS are shown in Figure 20a. More details on the DLTS/DLOS principle and deep-level defects in β-Ga_2_O_3_ are reported in ref. [247].

While DLTS/DLOS mainly characterize bulk traps, photo-assisted C-V (PCV) can be used for extracting deep-level interface and dielectric bulk traps at β-Ga_2_O_3_/dielectric interfaces. Two main PCV methods have been reported, where the first uses above-bandgap light and compares ΔV the dark and UV C-V curves vs. surface potential to calculate D_t_, which is the sum of the interface trap state density (D_it_) and the dielectric bulk trap density (n_bul_). An average D_it_ is found from the y-intercept of D_t_ vs. t_ox_ (Figure 20b). Note that the dark CV curve is measured from accumulation to depletion after being held in accumulation for 10 min for all interface traps to fill. In depletion, the device is exposed to UV light to excite electrons from all interface traps, and held in depletion for 10 min in the dark after UV exposure so that the generated holes move to the Al_2_O_3_/β-Ga_2_O_3_ interface [248]. The second method uses at least two sub-bandgap light sources to empty interface traps at two energies below E_C_. The resulting flatband voltage shift (ΔV_fb_) determines the D_it_. The second method has the advantage that no e–h pairs are generated, which can be a source of error, and a more precise D_it_ can be found as opposed to an average value [249].

**Figure 20 materials-16-07693-f020:**
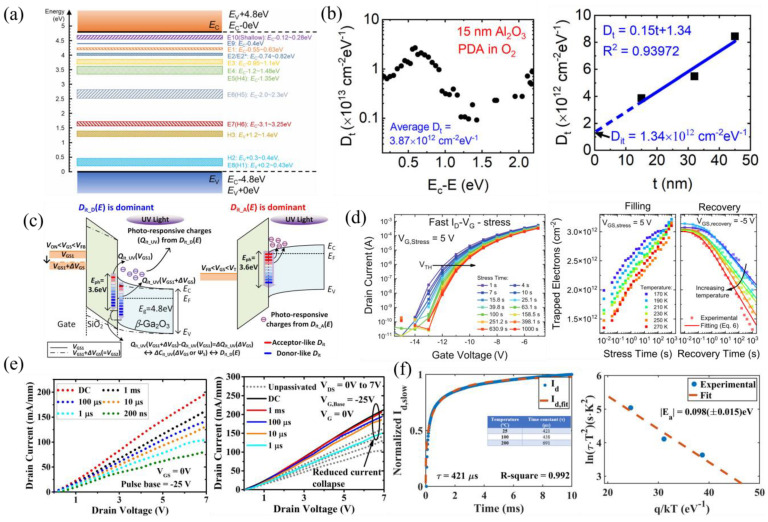
(**a**) Bulk trap levels found via DLTS and DLOS. Reproduced from [247]. © IOP Publishing. Reproduced with permission. All rights reserved. (**b**) PCV method using above-gap light for D_t_ and extrapolation to obtain an average D_it_. Reproduced from [248], with the permission of AIP Publishing. (**c**) Photo I-V to obtain donor and acceptor interface trap state densities. © (2018) IEEE. Reprinted with permission from [250]. (**d**) Stress I-V and trapped charge from stress C-V. Reproduced from [251], with the permission of AIP Publishing. (**e**) Pulsed I-V showing current dispersion (left) without passivation and significant improvement after SiN_x_ passivation (right). (**f**) Fitted I-V transient to variable range-hopping mechanism for time constant and temperature-dependent measurements for activation energy. © (2021) IEEE. Reprinted with permission from [133].

A photo I-V characterization method was developed to determine donor and acceptor interface trap state densities by comparing the subthreshold drain current in dark and UV conditions, attributing the range V_on_ < V_GS_ < V_fb_ due to donor traps and V_fb_ < V_GS_ < V_th_ due to acceptor traps [250].

Stress measurements are another trapping-characterization method, where stress I-V determines the V_th_ instability and stress C-V quantifies the trapped charge. A study of stress measurements on a β-Ga_2_O_3_ MOSFET observed a logarithmic dependence of V_th_ on stress time, as well as full recovery after 365 nm of UV illumination. The stress C-V at various temperatures also indicated that the trapping followed an inhibition model, where a trapped electron inhibits neighboring charge trapping due to coulombic repulsion [251]. Monitoring V_th_ shifts and R_on_ under positive bias stress (PBS) and negative bias stress (NBS) can help to identify traps contributing to the FET instability and degradation. This has been studied in recessed-gate [252], p-NiO-gate [253], and β-Ga_2_O_3_/SiC FETs [254]. PBS-induced instability is primarily caused by border traps in the gate oxide, while NBS-induced instability is attributed to both interface states and border traps [252]. In p-NiO gated FETs, a high V_GS_ or long stress time permanently negatively shifted the V_th_ by ionizing interface dipoles, which neutralized ionized charges in the depletion region [253]. The PBS of β-Ga_2_O_3_/SiC FETs fabricated via H^+^ ion implantation, as described in [156], positively shifted the V_th_ for short stress times due to electron trapping by the interface and border traps. For longer stress times, however, negative V_th_ shifts and an increase in carrier concentration were observed and attributed to the generation of shallow donors by H^+^ interstitials or the H passivation of deep-acceptor gallium vacancies [254].

Pulsed I-V, studied in nearly all material systems including SiC and GaN, is useful in that it can isolate the effects of buffer traps (in the case of drain pulsing) and surface/interface traps (in the case of gate pulsing) [255]. In some β-Ga_2_O_3_ studies, the drain lag had no DC-RF dispersion, while the gate lag exhibited significant current dispersion, indicating that surface traps near the β-Ga_2_O_3_/Al_2_O_3_ interface under the gate and in the drain–gate access region heavily impacted the RF performance, while buffer traps had minimal effects [96,133]. SiN_x_ passivation is seen to improve current dispersion in gate-lag measurements. Additionally, the fitting of the drain current transient provides information on the trap time response and trapping/de-trapping mechanism (Shockley–Read–Hall, variable range hopping, etc.), and temperature-dependent pulsed measurements can give trap activation energies.

#### 4.1.2. Material Preparation

It is widely known in β-Ga_2_O_3_ that there is Si contamination at the substrate/epi interface (Figure 21a), which can create a parasitic secondary channel, adding parasitic resistance and capacitance [216]. Additionally, the band bending at the substrate/epi interface depletes the channel [109], and semi-insulating impurities can diffuse into the channel region. Therefore, the growth of a buffer layer is useful in mitigating both the parasitic channels from Si contamination and channel depletion from the substrate. In delta-doped MESFETs, Fe-doped semi-insulating substrates reduced both the charge density and mobility with decreasing buffer thickness as well as increased RF dispersion (Figure 21b [103]). A secondary-ion mass spectroscopy (SIMS) depth profile showed that Fe impurities diffused nearly 200 nm into the epi layer, making the buffer thickness a critically important feature in lateral β-Ga_2_O_3_ FETs. To mitigate the effects of the second parasitic channel, a Mg delta-doping layer was grown at 420 °C using a 10 s open/30 s close/10 s open pulsing shutter scheme near the substrate/epi interface to compensate any Si impurity concentrations and reduce device leakage. An improved leakage current by six orders of magnitude and negligible hysteresis in transfer I-V curves was reported (Figure 21c [106]).

In another study, two buffer trap levels at E_C_—0.7 eV and E_C_—0.8 eV, associated with Fe-doped substrates, were observed using isothermal constant drain current DLTS (CI_D_-DLTS) (Figure 21d). The former was observed in MESFETs with buffer layers of both 100 nm and 600 nm, while the latter was not seen in the 600 nm buffer layer MESFET, leading to the conclusion that the trap at E_C_—0.8 eV is correlated with Fe diffusion into the buffer layer, while the trap at E_C_—0.7 eV is consistent with a point defect source observed in β-Ga_2_O_3_. The RF dispersion was not significantly reduced and an increased V_th_ variation was observed in the MESFET with the 600 nm buffer, indicating that the E_C_—0.7 eV trap was dominant in R_on_ degradation and V_th_ instability [105].

Techniques to reduce defect densities and their effects on device performance include pre-dielectric deposition cleans, post-deposition (PDA) and post-metallization annealing (PMA), in situ dielectric growth, and MacEtch device fabrication. Piranha treatment has shown to reduce the surface roughness and interface trapped charge, Q_it_, from 1.4 × 10^12^ cm^−2^ to 3.2 × 10^11^ cm^−2^. PDA at 500 °C in O_2_ or N_2_/O_2_ after piranha treatment showed a highly improved interface quality and lowered the average D_it_ to 2.3 × 10^11^ cm^−2^ eV^−1^ (O_2_ PDA) [256]. A study of PDA and PMA temperatures on interface quality observed that PMA at 300 °C in N_2_ after low-temperature PDA from 300 °C to 600 °C shifted the V_fb_ to near-ideal values and fixed the charge to the order of 1 × 10^11^ cm^−2^; however, PMA had little effect when the PDA temperature was increased from 700 to 900 °C due to Ga and Al interdiffusion. PMA significantly reduced the shallow D_it_ states for all PDA temperatures (lowest for PDA at 300 °C), but had no effect on the deep D_it_. The deep D_it_ states decreased only with increasing PDA from 300 to 900 °C, in contrast to the increasing density of shallow D_it_ states with increasing PDA temperature (Figure 22a [249]). Islam et al. recently reported on a solvent (S), O_2_ plasma, and piranha (P) surface-cleaning method, followed by BHF (B) surface etch, PE-ALD Al_2_O_3_, and in situ forming-gas PDA (FG-PDA) at 250 °C to achieve a ΔV_fb_ of 300 mV and 80 mV for the first and second C-V cycle, respectively, as well as stable accumulation at positive V_G_, while other comparative samples showed large first- and/or second-cycle hysteresis and no accumulation (Figure 22b [228]). Another study on surface damage removal after ICP used post-tetramethyl ammonium hydroxide (TMAH) and self-reaction etching (SRE) using Ga flux in an MBE chamber at 900 °C, after an observed reduced and negative growth rate with increased Ga flux, reducing D_it_ to 7.3 × 10^11^ cm^−2^ eV^−1^ [257]. The SRE method removed surface damage and improved C-V characteristics (Figure 22c). SRE can be useful when performing in situ gate dielectric growth, as initially reported by another group, achieving high breakdown fields of 5.8 MV cm^−1^ and an average D_it_ of 6.4 × 10^11^ cm^−2^ eV^−1^ [226].

MacEtch is not quite a material preparation method, but an alternative to FinFET fabrication that avoids dry-etch-induced damage, with hysteresis of only 9.7 mV and SS of 87.2 mV dec^−1^ [125].

A negligible hysteresis of 4 mV, µs switching, and a near-record low SS of 66 mV dec^−1^ was recently achieved in recessed p-NiO_x_-gated FETs using only piranha surface treatment [75]. This provides a potential path for maximizing β-Ga_2_O_3_ FET performance while maintaining ultra-low interface defect densities.

## 5. Current Challenges and Major Strategies

### 5.1. Lack of p-Type Doping

The unavailability of shallow acceptors in β-Ga_2_O_3_ prevents the fabrication of homoepitaxy pn diodes, guard rings, and superjunction devices. Heterojunction devices with p-type materials such p-NiO, p-GaN, p-SnO, and p-CuO_2_ have been investigated with some of the highest reported BFOMs, as discussed in the previous sections. Additionally, the interface properties in p-NiO-gate FETs reveal no hysteresis, low SS, and large mobility, indicating a high-quality interface and minimal trap scattering. While p-NiO is currently the most promising of the p-type materials for heterojunctions, its deposition via sputtering and polycrystallinity can result in nonuniformity and a low yield, which needs further investigation.

Some groups have reported p-doping using amphoteric Zn and H diffusion (see Section 2.5); however, such devices have yet to be reported. The feasibility of these techniques in high-performing diodes and FETs need to be verified for these techniques to be readily adopted in the high-power and RF markets.

### 5.2. Low Thermal Conductivity

The low thermal conductivity is a major issue in β-Ga_2_O_3_ devices, especially for high-power applications where self-heating is unavoidable. Various thermal studies using high-thermally conductive substrates have been reported and are discussed in the GO/AlN, GO/SiC, and GO/Diamond sections, and illustrated in Figure 16. There seem to be promising solutions using the hetero-integration of thermally conductive substrates via SiC ion-cutting technology or via flip-chip to a diamond carrier, thermal bumps, and NCD passivation.

### 5.3. Monolithic and Heterogeneous Integration

So far, most β-Ga_2_O_3_ devices have been stand-alone; however, it is imperative to incorporate them within circuitry to fully realize the potential of β-Ga_2_O_3_. Monolithic integration refers to circuitry designed on the same sample, which has so far been demonstrated with an inverter with SOI D-/E-mode graphene-gated FETs. The amplifier capability of RF FETs, obtained by determining their gain, output power, and power-added-efficiency using CW power measurements, highlights the utility of these FETs in integrated circuits (Table 4). Heterogeneous integration has mainly been realized with SOI FETs on high-thermally conductive substrates.

While the mechanical exfoliation of β-Ga_2_O_3_ nanomembranes is used in most devices, it is best-suited for proof of concept, and not for large-scale production. For β-Ga_2_O_3_, two methods of wafer-to-wafer bonding have been demonstrated, including ion cutting using H^+^ implantation, tested on SiC and Si substrates, [156] and low-temperature fusion bonding on SiC using a SiN_x_ interlayer [201]. Another heterogeneous integration method for improved temperature reduction is flip-chip bonding to a diamond carrier, with the downside that, currently, diamond cannot be supplied in large wafers [204].

### 5.4. Packaging

Packaging is essentially the next step after device-level thermal management. Experimentally verified cooling methods need to translate accordingly for large-area, packaged devices. Additionally, both the device-level and package-level thermal management must be co-designed. One limitation is that device software and package software are difficult to integrate because each simplifies the results of the other [258].

### 5.5. Optical Effects and Remote Switching

The large, near-direct bandgap of β-Ga_2_O_3_ creates potential for solar-blind deep-UV (DUV) photodetectors, which is an ongoing area of research in β-Ga_2_O_3_-based devices [6,259,260,261]. The absorption anisotropy due to the complex crystal structure [262], as well as the strong sub-gap absorption from the conduction band to oxygen and gallium vacancies [263], remains a challenge for DUV β-Ga_2_O_3_ photodetectors. The optical characteristics of β-Ga_2_O_3_ may have potential advantages in the remote switching of high-power RF amplifier circuits. Remote switching is a cost-effective technique that can improve switching speeds while reducing or eliminating electrical noise. This has been discussed in GaN-based systems [264,265,266,267] and could likewise apply in β-Ga_2_O_3_-based systems.

### 5.6. Requirements in Real Applications

In high-power applications, E_br_ and R_on,sp_ matter less than breakdown voltage and on-current. Small-area devices with high BFOMs should mainly be used as an in-between step to fabricate an equivalent large-area device to meet specific current and voltage ratings for real applications. This can provide further insight into what device optimizations are needed if the large-area device underperforms.

## 6. Applications and Trends

β-Ga_2_O_3_ FETs are not expected to replace their SiC and GaN counterparts because they are already commercially available. While this might occur in the future, the current trend is that high-power β-Ga_2_O_3_ FETs, with voltage and current ratings beyond those of GaN and SiC devices, will be used for ultra-high-power applications such as electric vehicles, rails, power grids, renewable energy storage, etc. High-power RF FETs can also find use in electric vehicles, power converters, data centers, and communication applications. The emergence of β-Ga_2_O_3_ RF FETs is difficult, with their performance still being below that of GaN HEMTs and newer diamond HEMTs. However, the low-cost melt growth technique, compared to the high cost of diamond, as well as the higher theoretical v_sat_ than GaN, both show a promising future for high-frequency β-Ga_2_O_3_ devices. High-power β-Ga_2_O_3_ FETs have shown higher breakdown fields surpassing the GaN theoretical limit, and so a potentially more accessible market for RF β-Ga_2_O_3_ FETs is mid-frequency (≈tens of GHz), high-power RF FETs that can outperform high-power GaN RF FETs.

## 7. Conclusions and Outlook

In conclusion, considerable advancements in β-Ga_2_O_3_ FET designs have been made to push both their high-power and RF capabilities. High-power FETs have shown breakdown voltages up to 10 kV, current densities of >1 kA cm^−2^ and 1.5 mA mm^−1^, and BFOMs of 0.95 GW cm^−2^. RF FETs have reported high breakdown fields of 5.4 MV cm^−1^, operating frequencies up to 48 GHz f_max_, and saturation velocities up to 3 × 10^6^ cm s^−1^. While many FETs have surpassed the theoretical FOMs of Si, they are still far from those of β-Ga_2_O_3_. From the overview of β-Ga_2_O_3_ FETs, the key takeaways are as follows:(1)The importance of high-quality epitaxial growth and buffer layers cannot be understated. The highest BFOM FET to date also reports one of the highest mobilities of 184 cm^2^ V^−1^ s^−1^, realized through MOCVD varied low/high temperature layers;(2)The SAG is vital to both high power and RF in that it is used to scale device geometries and reduce source–gate series resistance. It should be implemented, if possible, in both lateral and vertical FETs;(3)For high currents, vertical transistors are preferred because the current scales with the device area as opposed to the channel thickness, as in lateral devices. FinFETs and CAVETs show the best results, with FinFETs offering more gate control and reduced leakage, but increased complexity. MacEtch FinFETs are a non-dry-etching alternative;(4)Normally off (E-mode) FETs are crucial for power electronics because of their reduced off-state power loss, fail-safe high-voltage operation, and simplified circuitry for power switching. The lack of p-type doping, and therefore inversion, in β-Ga_2_O_3_ requires approaches such as recessed gates (Section 3.1.3), low-doped channels and CBLs (Section 3.2.1), small-width FinFETs (Section 3.1.4), oxygen annealing (Section 3.2.2), and p-gate materials for normally off operation (Section 3.4.2);(5)FP structures (GFP, SFP) including T-gates are vital to any high-power device. High-k or extreme-k FP dielectrics are an attractive option to improve breakdown;(6)SOI FETs can be very useful in conducting studies on thermal, transport, novel gate dielectrics, etc. However, they are limited in their breakdown voltage and small sample size. SOI FETs should be considered as a proof of concept with the intent to apply successful designs into bulk devices;(7)Novel structures simulated through TCAD, such as vertical trench gates, GAA, air-gap FPs, HBTs, and others, should be used to evaluate the potential of a design before fabrication;(8)RF FETs have been realized in delta-doped MESFETs, AlGO/GO MODFETs, and HFETs, forming a 2DEG with Si-doped AlGO/UID-GO;(9)One commonality of RF FETs is their T-gate structure, allowing highly scaled L_G_ while maintaining low noise figures;(10)RF FETs have reported high operating frequencies at ≥27 GHz with and without FP dielectrics;(11)Ohmic contacts should always utilize some improvements, such as regrowth, ion implantation, or interlayers;(12)P-NiO-gate dielectrics show promise in increasing the BFOM, while maintaining high-quality/low-defect density interfaces. A high-bandgap dielectric should be added to increase the gate swing beyond the pn turn-on voltage;(13)Thermal management is crucial, and the intention to use wafer-bonding techniques or flip-chips with high-thermally conductive substrates must be implemented to further enhance device performance;(14)For high-power applications, an appealing FET design with high FOM(s) should be fabricated as a large-area device to meet current and breakdown voltage ratings.

Defect characterization is vitally important in β-Ga_2_O_3_, and techniques need to be adapted or invented for UWBG materials. Material preparation is essential to improve peak performance and must be considered in each step of fabrication.

Device-level and package-level thermal management and modeling is critical in taking β-Ga_2_O_3_ devices to the market. Thermally conductive substrates have experimentally shown significant drops in peak temperatures, and modeling has indicated that wafer bonding via flip-chip and junction-side cooling can reduce thermal effects. The device and package must be designed and optimized simultaneously; however, co-design modeling is still limited.

Overall, the rapid progress in material quality, fabrication, defect characterization and mitigation, and thermal management indicates tremendous potential for β-Ga_2_O_3_ devices to quickly enter the power electronics application space once the presented challenges are addressed.

## Figures and Tables

**Figure 1 materials-16-07693-f001:**
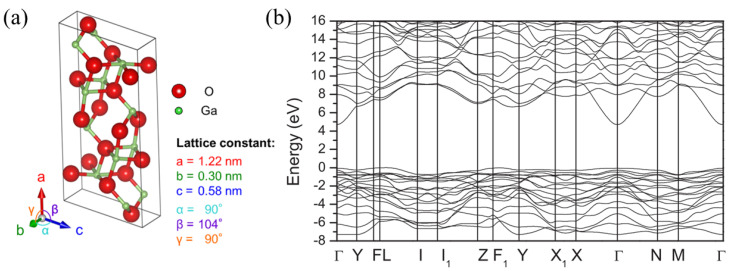
(**a**) β-Ga_2_O_3_ unit cell. Reproduced from [22]. © IOP Publishing. Reproduced with permission. All rights reserved. (**b**) β-Ga_2_O_3_ band diagram. Reprinted with permission from [19]. Copyright 2017 by the American Physical Society.

**Figure 2 materials-16-07693-f002:**
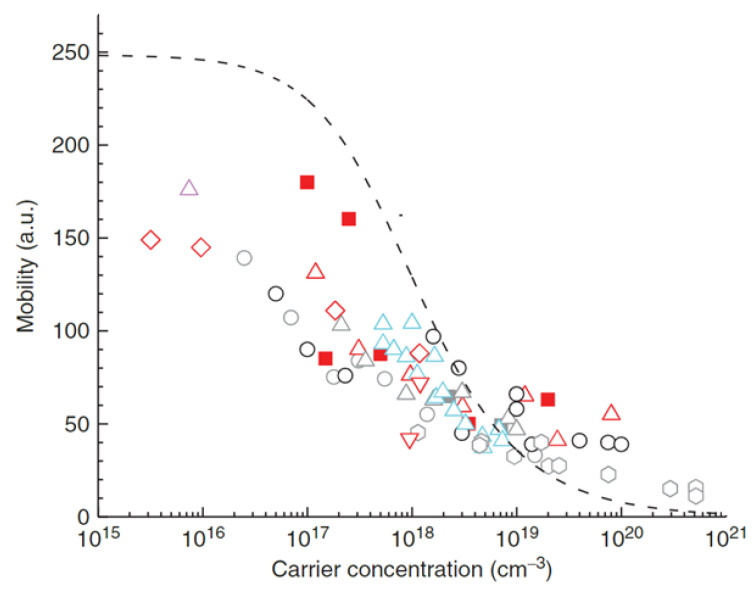
Electron mobility vs. carrier concentration in β-Ga_2_O_3_ for Si, Sn, and Ge dopants in layers grown using various crystal and thin film techniques. Adapted with permission from Chen et al. [31] © 2023 John Wiley & Sons, Ltd.

**Figure 3 materials-16-07693-f003:**
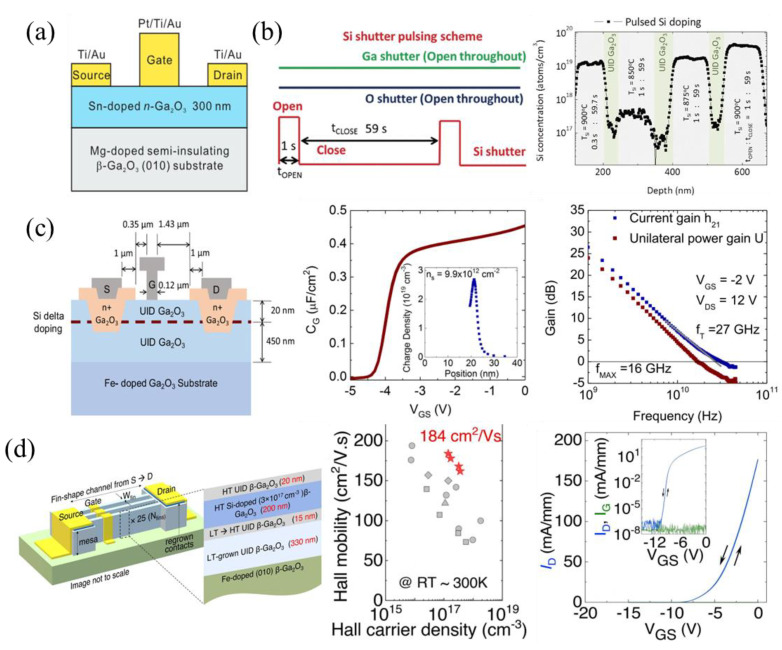
(**a**) First MESFET reported in 2013. Reproduced from [101], with the permission of AIP Publishing. (**b**) Shutter pulsing scheme and doping variation showing alternating UID and uniformly doped layers. Reproduced from [102]. © The Japan Society of Applied Physics. Reproduced with the permission of IOP Publishing Ltd. All rights reserved. (**c**) Highly scaled T-gate delta-doped MESFET with high cutoff and maximum frequencies. © (2019) IEEE. Reprinted with permission from [88]. (**d**) Tri-gate MESFETs with low-temp/high-temp grown layers resulting in ultra-high mobilities and negligible I-V hysteresis. © (2022) IEEE. Reprinted with permission from [63].

**Figure 4 materials-16-07693-f004:**
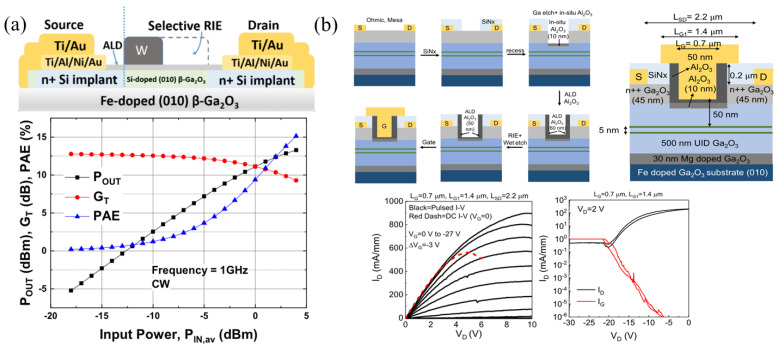
(**a**) SAG FET using refractory metal gate W and Si ion implantation for self-alignment with an L_SG_ of 0 µm. The RF P_out_, G_T_, and PAE as a function of input power at 1 GHz are plotted. Reproduced from [89]. CC BY 4.0. (**b**) SAG process enabled by using an n++ grown cap layer as opposed to ion implantation. High gate leakage and low on/off ratio are indicative of a leaky dielectric due to its deposition or residual Ga droplets at the interface. Reproduced from [61], with the permission of AIP Publishing.

**Figure 6 materials-16-07693-f006:**
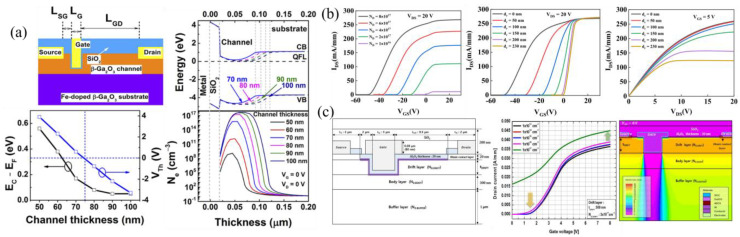
(**a**) TCAD model of a recessed-gate FET studying variations in fermi level, V_th_, and electron concentration with channel thickness. Reproduced from [115]; licensed under a Creative Commons Attribution (CC BY) license. (**b**) TCAD model of a recessed-gate FET studying variations in V_th_ and current density with doping and recess depth. Reprinted from [116], Copyright (2023), with permission from Elsevier. (**c**) A novel recessed-gate FET design with different body and drift layers recessing fully through the drift layer. Body-doping effects on E-/D-mode operation as well as a 2D cross-section of band bending through the body layer at low dopings. Reproduced from [117]. CC BY 4.0.

**Figure 7 materials-16-07693-f007:**
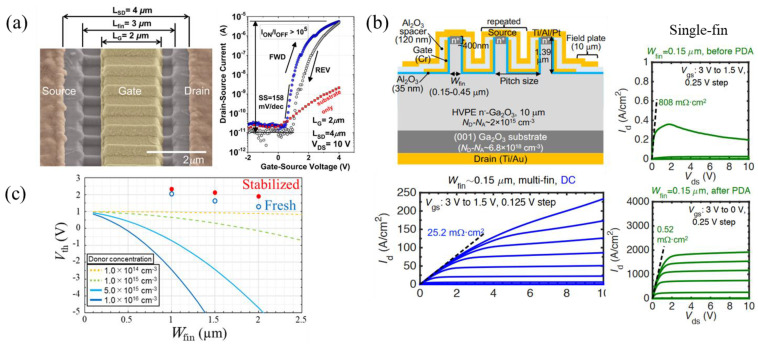
(**a**) Lateral E-mode FinFETs and transfer curves with observed substrate conduction due to free carriers at the semi-insulating substrate. Reproduced from [113]. CC BY 4.0. (**b**) Cross-section of vertical multi-fin FETs, I-V curves of a single-fin FET showing significant improvement through PDA, and multi-fin FET I-V curves. © (2019) IEEE. Reprinted with permission from [81]. (**c**) Nitrogen doping mitigating the V_th_ dependence on W_fin_ and maintaining E-mode operation for large W_fin_. Reproduced from [121]. © The Japan Society of Applied Physics. Reproduced by permission of IOP Publishing Ltd. All rights reserved.

**Figure 8 materials-16-07693-f008:**
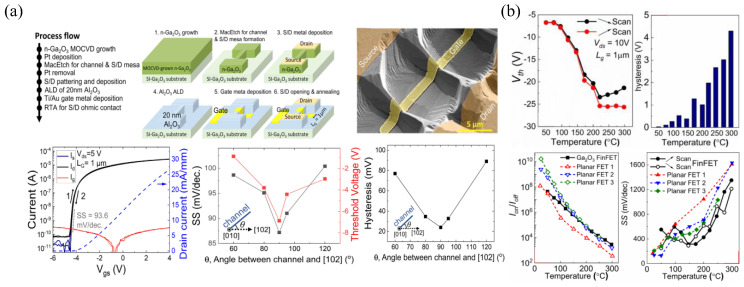
(**a**) FinFETs fabricated via MacEtch with process and TEM image. I-V curves as well as SS and hysteresis dependence on channel angle relative to [102] are shown. Channels perpendicular to [102] show the best performance. Reproduced from [125], with the permission of AIP Publishing. (**b**) Temperature dependence of V_th_, hysteresis, on/off ratio, and SS in MacEtch FinFETs indicating thermal degradation of the interface and/or dielectric. Reprinted from [127], with the permission of AIP Publishing.

**Figure 10 materials-16-07693-f010:**
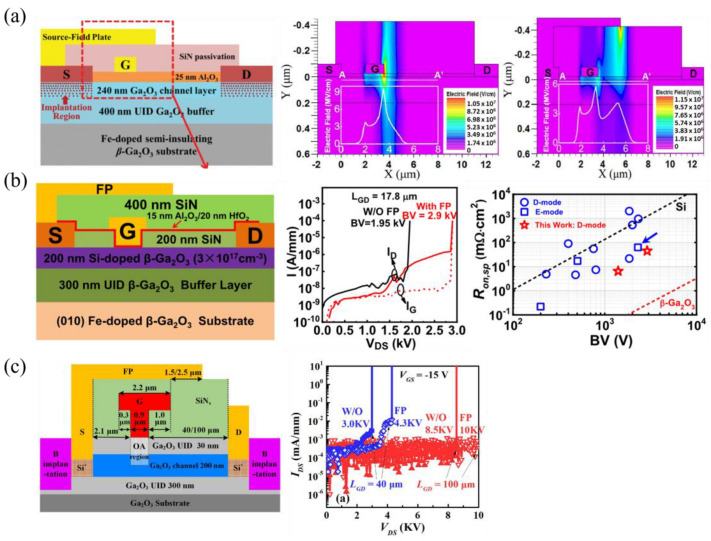
(**a**) Lateral MOSFET with SFP and simulated TCAD electric field profiles clearly showing field spreading and reduction in overall peak field value. © (2019) IEEE. Reprinted with permission from [137]. (**b**) FET with SFP and T-gate structure, breakdown I-V, and benchmark plot. © (2020) IEEE. Reprinted with permission from [65]. (**c**) FET with SFP, T-gate, oxygen annealing (OA), and B-implantation for device isolation. The blue/red lines represent an L_GD_ of 40/100 µm and solid/open symbols represent without/with SFP. A breakdown of 10 kV is observed. © (2023) IEEE. Reprinted with permission from [59].

**Figure 11 materials-16-07693-f011:**
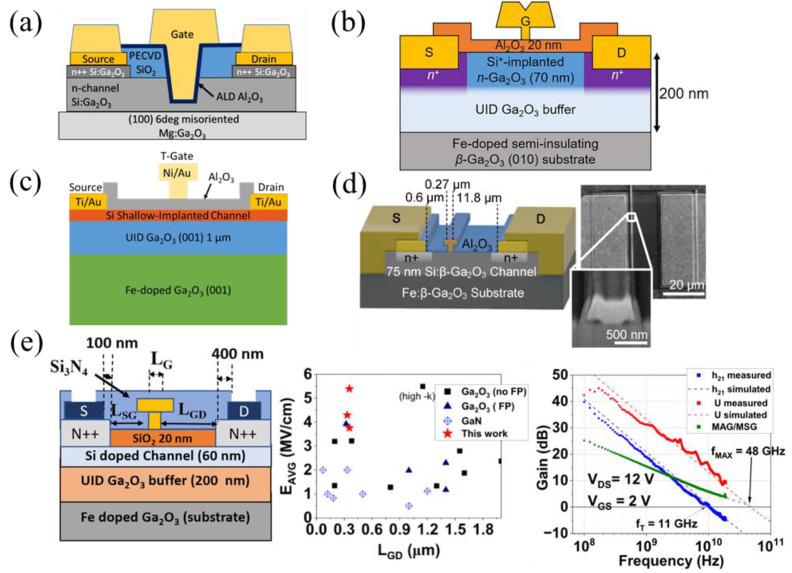
A variety of RF T-gate FETs are shown. (**a**) FET incorporates a recessed-gate architecture. Reprinted with permission from [139]. (**b**) FET uses air as the FP dielectric. Reprinted from [140], with the permission of AIP Publishing. (**c**) FET uses both an air FP dielectric and an ultra-thin implanted channel [94]. (**d**) FET incorporating Al_2_O_3_ surface and gate metal passivation. Reprinted from [66]. CC BY-NC-ND 4.0. (**e**) T-gate RF FET with SiN_x_ passivation with highest-reported f_max_ and high breakdown field of 48 GHz and 5.4 MV cm^−1^, respectively. Reproduced from [93], with the permission of AIP Publishing.

**Figure 12 materials-16-07693-f012:**
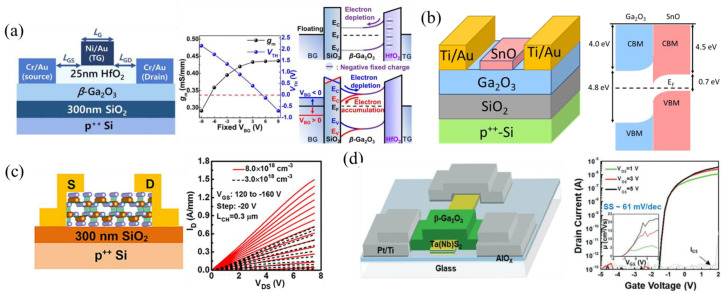
(**a**) SOI FET with V_th_ modulation using constant back-gate voltage to accumulate or deplete the channel, while the top gate controls the device. © (2019) IEEE. Reprinted with permission from [152]. (**b**) SOI FET obtaining a record mobility of 191 cm^2^ V^−1^ s^−1^ using a floating p-SnO layer in the channel [85]. (**c**) SOI FET with a p++ back gate and doping of 8 × 10^18^ cm^−3^, measuring record currents of 1.5 A mm^−1^. Reprinted from [68], with the permission of AIP Publishing. (**d**) TMD high Schottky barrier gate with near-ideal SS of 61 mV dec^−1^ when using TaS_2_. Reprinted with permission from Kim et al. [160] © 2023 Wiley-VCH GmbH.

**Figure 13 materials-16-07693-f013:**
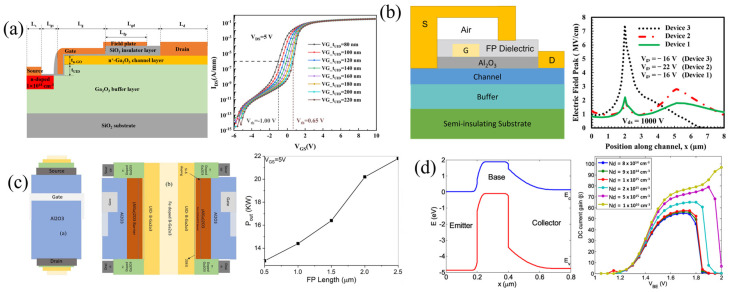
TCAD simulations of novel, not yet realized β-Ga_2_O_3_ FETs. (**a**) An FP self-aligned trench vertical gate with variable V_th_ based on gate trench thickness into the UID layer, t_UID_. © (2023) IEEE. Reprinted with permission from [166]. (**b**) SFP with air gap dielectric to better mitigate electric fields at device edges. Reprinted with permission from [168], Copyright Elsevier (2022). (**c**) GAA FET with 2DEG to improve P_out_ and f_T_. Reprinted with permission from [169], Copyright Elsevier (2021). (**d**) Band diagram and current gain of npn HBT using p-CuO_2_, but limited by p-oxide bandgap, interface traps, and CBO between emitter and base. Reproduced from [170]. © IOP Publishing. Reproduced with permission. All rights reserved.

**Figure 14 materials-16-07693-f014:**
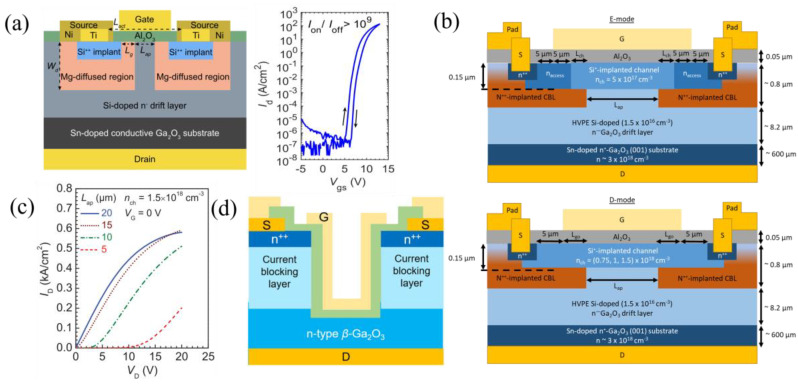
(**a**) Cross-section and transfer I-V curve of E-mode CAVET with CBL surrounding source. Copyright (2022) IEEE. Reprinted with permission from [83]. (**b**) E-/D-mode CAVETs implemented via n_ch_ variation [174]. (**c**) I-V output curves of CAVETs with different L_ap_ showing diode-like turn-on behavior. Reproduced from [176], with the permission of AIP Publishing. (**d**) U-MOSFET with CBL. Reproduced from [84], with the permission of AIP Publishing.

**Figure 15 materials-16-07693-f015:**
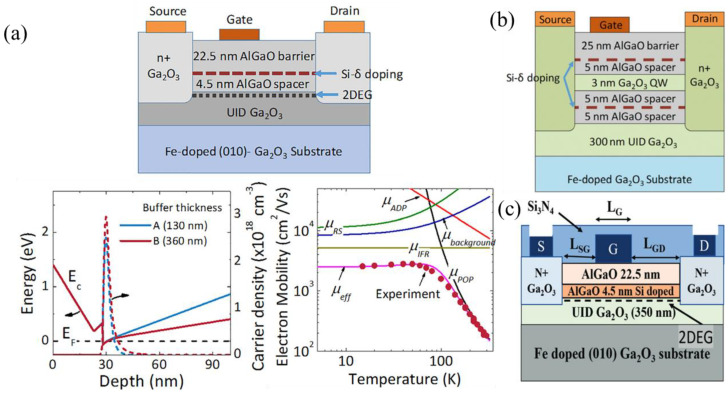
(**a**) Cross-section band diagram showing 2DEG and measured mobility up to 180 cm^2^ V^−1^ s^−1^ at room temperature of an AlGO/GO MODFET using delta doping. Reproduced from [187], with the permission of AIP Publishing. (**b**) Cross-section of a double-heterostructure MODFET. Reproduced from [189], with the permission of AIP Publishing. (**c**) Cross-section of a heterostructure FET. Reproduced from [96], with the permission of AIP Publishing.

**Figure 21 materials-16-07693-f021:**
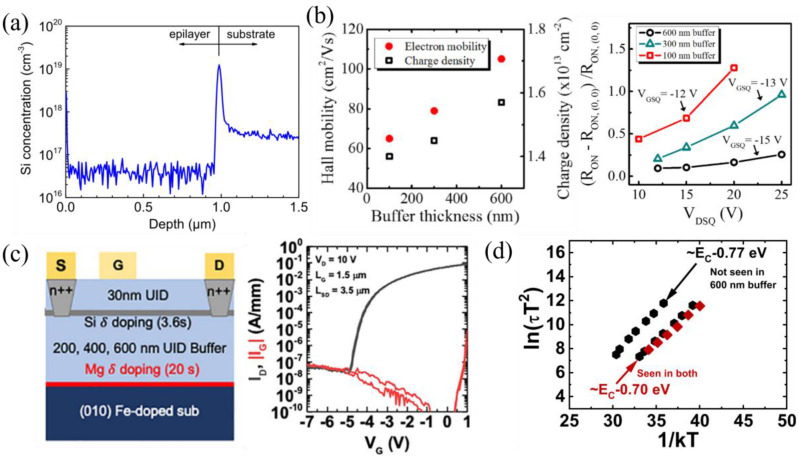
(**a**) Si concentration peak at the substrate/epi interface. Reproduced from [216]. © The Japan Society of Applied Physics. Reproduced with the permission of IOP Publishing Ltd. All rights reserved. (**b**) Negative effects of impurities from semi-insulating substrates and back depletion on mobility, charge density, and current dispersion. Reproduced from [103], with the permission of AIP Publishing. (**c**) Mg delta doping at substrate/epi interface to compensate Si impurities and reduce leakage current. Reproduced from [106], with the permission of AIP Publishing. (**d**) Buffer traps at E_C_—0.77 eV only for a 100 nm buffer, and at E_C_—0.70 eV for both a 100 nm and 600 nm buffer using CI_D_-DLTS. Reproduced from [105], with the permission of AIP Publishing.

**Figure 22 materials-16-07693-f022:**
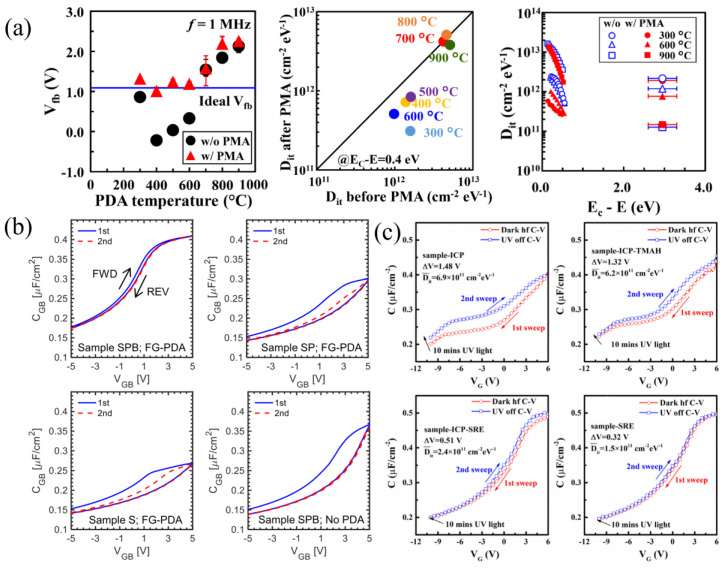
(**a**) Effects of PDA and PMA on V_fb_ and D_it_. PMA shows improvement in both reducing fixed charge and shallow D_it_, while having little effect on deep D_it_. Reproduced from [249], with the permission of AVS: Science & Technology of Materials, Interfaces, and Processing. (**b**) First and second C-V sweeps of MOSCAPs with different surface cleans. SPB with FG-PDA introduces the least interface defects relative to the others. Reprinted with permission from [228]. (**c**) PCV comparing SRE to TMAH for removing ICP damage. Reproduced from [257], with the permission of AIP Publishing.

**Table 2 materials-16-07693-t002:** Performance comparison of D-mode high-power FETs.

Ref.	FET Design	On/Off	I_D,max_ (mA mm^−1^)	V_br_ (V)	E_br_ (MV cm^−1^)	R_on,sp_ (Ω cm^2^)	µ (cm^2^ V^−1^ s^−1^)	BFOM (MW cm^−2^)
[59]	MESFET, T-gate + SFP, OA	10^6^	3.3	10 k	1	2.92	NR	>34.2
[60]	Delta-doped MESFET w/GFP	10^7^	180	315	2.3	NR	73	118
[61]	Delta-doped SAG	10^3^	560	NR	NR	NR	65	NR
[62]	Recessed and T-gate	10^9^	49	1.80 k	1.8	20.9 m	128	155
[63]	Tri-gate lateral FinFET	10^10^	187	1.13 k	4.2	1.34 m	184	950
[64]	Composite + SU8 GFP	10^9^	40	7.16 k	1.79	8.98	NR	5.71
[65]	SFP, T-gate, Al_2_O_3_/HfO_2_ gate oxide	10^9^	230	1.40 k	2.90	7.08 m	NR	277
[66]	Scaled T-gate MESFET	10^4^	60	2.45 k	2.08	17.3 m	84	347
[67]	SOI on sapphire	10^8^	232	800	NR	7.41 m	137	86.3
[68]	Back-gate SOI on SiO_2_/Si	10^10^	1500	NR	NR	NR	NR	NR
[69]	CAVET, N^++^ ion implant	10^8^	420 A cm^−2^	25	NR	31.5 m	140	NR
[70]	AlGO/GO w/GFP	10^8^	NR	1.37 k	0.86	120 m	101	15.6
[71]	SOI on AlN/Si	10^9^	580	118	1.04	1.44 m	82.9	9.70
[72]	SiC/GO composite wafer	10^8^	NR	2.37 k	1.23	18.4 m	94	303
[73]	SOI on Diamond	NR	980	NR	NR	NR	NR	NR
[74]	p-NiO gate oxide	10^10^	450	1.12 k	2.48	3.19 m	NR	390
[75]	p-NiO gate oxide	10^10^	282	2.15 k	3.5	6.24 m	130	740
[76]	p-NiO/SiO_2_ gate oxide	10^9^	300	1.32 k	1.47	4.30 m	NR	405
[77]	p-SnO gate oxide	10^6^	100	750	1.9	3.15 m	100	178
[78]	BTO (ε≈235) gate oxide	10^5^	359	640	1.5	1.08 m	72	376
[79]	Al_2_O_3_/BTO gate oxide	10^7^	220	840	4.10	1.72 m	85	408

**Table 3 materials-16-07693-t003:** Performance comparison of E-mode high-power FETs.

Ref.	FET Design	On/Off	I_D,max_ (mA mm^−1^)	V_br_ (V)	E_br_ (MV cm^−1^)	R_on,sp_ (Ω cm^2^)	µ (cm^2^ V^−1^ s^−1^)	BFOM (MW cm^−2^)
[80]	Recessed gate	10^9^	40	505	0.84	17.2 m	106	14.8
[81]	Multi-fin vertical FET	10^8^	230 A cm^−2^	2.66 k	NR	25.2 m	40	280
[82]	SOI on SiO_2_/Si	10^10^	450	185	2	20 Ω mm	55.2	NR
[83]	Mg-diffused CAVET	10^9^	150 A cm^−2^	72	NR	NR	7.5	NR
[84]	Vertical U-trench w/CBL	6.4 × 10^4^	11 A cm^−2^	102	NR	1.48	NR	0.007
[76]	p-NiO/SiO_2_ gate oxide	10^8^	NR	2.96 k	0.985	115 m	NR	76
[75]	p-NiO gate oxide	10^7^	43.2	1.98 k	3.3	13.8 m	140	284
[85]	Back-gate SOI on SiO_2_/Si p-SnO on top	2.26 × 10^6^	14.1	NR	NR	NR	191	NR
[86]	SOI on SiO_2_/Si HfO_2_ gate oxide	10^5^	11.1	80	0.16	82 m	81	0.078
[87]	Multi-stack gate: HZO/Al_2_O_3_/HfO_2_/Al_2_O_3_	10^8^	23.2	2.14 k	3.45	24 m	97	193

**Table 4 materials-16-07693-t004:** Performance comparison of D-/E-mode RF FETs.

Ref.	Type	Structure	On/Off	I_D,max_ (mA mm^−1^)	V_br_ (V)	E_br_ (MV cm^−1^)	µ (cm^2^ V^−1^ s^−1^)	f_T_ (GHz)	f_max_ (GHz)	G_p_ (dB)	G_T_ (dB)	P_out_ (mW mm^−1^)	PAE (%)	f_T_ V_br_ (THz V)	v_sat_ (f_T_ L_g_ 2π) (cm s^−1^)
[88]	D-M	T-gate delta-doped MESFET	108	260	150	1.07	70	27	16	NR	NR	NR	NR	4.05	2.01 × 10^6^
[89]	D-M	SAG	108	Pulsed ≈ 300	NR	NR	74	NR	NR	NR	13	715	23.4	NR	NR
[90]	D-M	Recessed gate SiO_2_ passivation	106	150	NR	NR	96	3.3	12.9	5.1	1.8	230	6.3	NR	1.45 × 10^6^
[91]	D-M	Tri-gate FinFET	NR	88	NR	NR	NR	5.4	11.4	NR	NR	NR	NR	NR	1.19 × 10^6^
[92]	D-M	SiO_2_ GFP	NR	58	NR	NR	NR	NR	NR	4.81	NR	130	22.4	NR	NR
[93]	D-M	T-gate, SiN_x_ passivation SiO_2_ gate oxide	1.23 × 10^5^	285	192	5.4	80	11	48	NR	NR	NR	NR	2.112	2.45 × 10^6^
[94]	D-M	T-gate, shallow ion-implanted channel	108	165	193	2.09	23	29	35	7	NR	11.2 dBm	11.6	5.597	2.73 × 10^6^
[95]	D-M	OA, SiN_x_ T-gate Multi-stack gate oxide: Al_2_O_3_/HfO_2_	109	200	NR	NR	75	1.8	4.2	3.6	NR	430	6.42	NR	1.13 × 10^6^
[96]	D-M	AlGO/GO HFET	NR	Pulsed ≈ 80	NR	NR	NR	14	22	NR	NR	NR	NR	NR	1.76 × 10^6^
[97]	E-M	AlGO/GO HFET	1.55 × 10^5^	74	23	1.35	NR	30	37	NR	NR	NR	NR	NR	3.02 × 10^6^
GaN and Diamond RF FETs															
[98]	D-M	GaN HEMT	10^3^	1000	60	0.4	1900	104	205	8	NR	5100	43.6	6.24	9.80 × 10^6^
[99]	D-M	GaN HEMT	3 × 10^5^	Pulsed 1300	50	NR	1423	156	308	15	NR	2500	70	7.8	5.89 × 10^6^
[100]	D-M	Diamond HEMT	NR	500	121	0.81	101	6.2	17	12.2	NR	4200	21.5	0.75	3.51 × 10^6^

## Data Availability

No new data were created or analyzed in this study. Data sharing is not applicable to this article.

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
