# Peer review of "Progress in Gallium Oxide Field-Effect Transistors for High-Power and RF Applications"

_materials, 2023, doi:10.3390/ma16247693_

Round 1

Reviewer 1 Report

Comments and Suggestions for Authors

This contribution reports on the recent progress of FET transistors for power and RF applications.

In my opinion the topic is of interest for the present journal.

The discussion is very in-depth and the article is well-written and organized.

Also the English level is satisfactory.

Although the authors mentioned almost all the most relevant topics, different researchers in the literature addressed the topic of the effects of an optical signal on the behavior of GaN high frequency HEMT. In my opinion, such a complete review should at least mention this research field, indeed it can be exploited as an example to remotely switch on an amplifier. I reported some references, hereafter.

Caddemi, A. et al., Microwave effects of UV light exposure of a GaN HEMT: Measurements and model extraction, Microelectronics Reliability, Volume 65, 2016, Pages 310-317, doi: 10.1016/j.microrel.2016.08.020

Caddemi, A.; et al. Light Exposure Effects on the DC Kink of AlGaN/GaN HEMTs. Electronics 2019, 8, 698. doi: 10.3390/electronics8060698.

Liang, Y.; et al. The study of the contribution of the surface and bulk traps to the dynamic Rdson in AlGaN/GaN HEMT by light illumination. Appl. Phys. Lett. 2016, 109, 182103–182106.

I really appreciated the section focused on the current challenges.

However, I suggest to add an additional section before the conclusion with the task to summarize the most promising applications and trends of this technology.

Reviewer 2 Report

Comments and Suggestions for Authors

This is an excellent and comprehensive review paper on gallium oxide power and RF field-effect transistor. The authors systemically reviewed the structure, growth and doping of Ga2O3, Ohmic contacts, gate materials, Schottky FETs and MOSFETs. The heterostructures based don Ga2O3 were also presented. The paper is well written and organized. I recommended the acceptance of this paper. I have only minor comments.

(1) There are already several reviewer paper (i.e. doi: 10.1088/1674-4926/44/6/061801) although yours is the newest. Such kind of papers should be referred and the author should show the difference between yours and others briefly in the introduction.

(2) Normally off operation is very important for power electronics. The authors may briefly discuss this point in the conclusion by citing the latest literatures.
(3) The bias stress induced stability can be discussed a little bit more with the latest literatures (i.e. doi: 10.1088/1674-4926/44/7/072803. ).
(4) Although the RF performance of Ga2O3 FET may be worse than those of HEMTs and diamond FETs. A table is recommended to compare the FETs of G2O3 with emerging diamond (doi: 10.1080/26941112.2022.2082853) as well as well developed HEMTs with regard to the power density and cut-off frequency, etc.
